

# The anatomy and phylogenetic position of the erythrosuchid archosauriform *Guchengosuchus shiguaiensis* from the earliest Middle Triassic of China

Richard J. Butler[1], Martín D. Ezcurra[1,2], Jun Liu[3,4], Roland B. Sookias[5] and Corwin Sullivan[6,7]

[1] School of Geography, Earth and Environmental Sciences, University of Birmingham, Birmingham, UK
[2] Sección Paleontología de Vertebrados, CONICET–Museo Argentino de Ciencias Naturales "Bernardino Rivadavia", Buenos Aires, Argentina
[3] Key Laboratory of Vertebrate Evolution and Human Origins of Chinese Academy of Sciences, Institute of Vertebrate Paleontology and Paleoanthropology, Chinese Academy of Sciences, Beijing, China
[4] CAS Center for Excellence in Life and Paleoenvironment, Beijing, China
[5] Museum für Naturkunde, Leibniz Institute for Evolution and Biodiversity Science, Berlin, Germany
[6] Department of Biological Sciences, University of Alberta, Edmonton, Canada
[7] Philip J. Currie Dinosaur Museum, Wembley, Canada

Corresponding author
Richard J. Butler,
r.butler.1@bham.ac.uk

## ABSTRACT

Erythrosuchidae is a clade of early archosauriform reptiles, which were apex predators in many late Early and Middle Triassic ecosystems, following the Permo-Triassic mass extinction. Erythrosuchids had a worldwide distribution, with well-preserved fossil material known from South Africa, European Russia, and China. We here redescribe the anatomy and revise the taxonomy of *Guchengosuchus shiguaiensis*, which is one of the stratigraphically oldest erythrosuchids and is known from a single partial skeleton from the lowermost Middle Triassic (lower Anisian) lower Ermaying Formation of Shaanxi Province, China. We provide a new differential diagnosis for *Guchengosuchus shiguaiensis*, and identify a series of autapomorphies relating to the morphologies of the skull roof and vertebrae. Incorporating updated anatomical information for *Guchengosuchus* into the most comprehensive morphological phylogenetic analysis available for early archosauromorphs recovers it as an early branching member of Erythrosuchidae, outside of the clade formed by *Garjainia*, *Erythrosuchus*, *Chalishevia*, and *Shansisuchus*. *Fugusuchus hejiapanensis*, from the uppermost Lower Triassic to lower Middle Triassic Heshanggou Formation of China, is recovered as the earliest branching member of Erythrosuchidae.

## INTRODUCTION

Erythrosuchidae is a clade of early archosauriform reptiles that comprises a small number of species ranging stratigraphically from the late Early to the Middle Triassic

(*Ezcurra, Butler & Gower, 2013*; *Ezcurra, 2016*). Erythrosuchids were important apex predators in earliest Mesozoic ecosystems, and are characterized by their proportionately large skulls and hypercarnivorous adaptations (*Ezcurra, Butler & Gower, 2013*). Although the taxonomic content, phylogenetic position, and interrelationships of Erythrosuchidae have long been unclear, substantial work over the last two decades has greatly increased understanding of the group (*Gower, 1997*, *2003*; *Ezcurra, Butler & Gower, 2013*; *Wang et al., 2013*; *Gower et al., 2014*; *Ezcurra, 2016*; *Ezcurra et al., 2018*), and demonstrated a geographic distribution including South Africa (*Gower, 2003*; *Gower et al., 2014*), Russia (*Ochev, 1958*; *Huene, 1960*; *Gower & Sennikov, 2000*; *Ezcurra et al., 2018*), China (*Wang et al., 2013*; *Ezcurra, 2016*), India (*Bandyopadhyay, 1999*), and possibly Australia (*Ezcurra, 2016*). However, the anatomy and taxonomy of several species within the clade remain poorly understood.

One of the least well–understood erythrosuchids is *Guchengosuchus shiguaiensis* from the earliest Middle Triassic of Shaanxi, China (Fig. 1). *Peng (1991)* provided a brief description in Chinese of *Guchengosuchus*, with a relatively small number of figures, but this taxon has received little subsequent attention and was not included by *Parrish (1992)* in his analysis of the phylogeny of Erythrosuchidae. Indeed, the first inclusion of this species in a quantitative phylogenetic analysis was by *Ezcurra (2016)*, who recovered it as the earliest branching member of the erythrosuchid clade, making it potentially significant for understanding the origins of the distinctive body plan of the group. Here, we provide a full redescription of the anatomy of *Guchengosuchus*, revise its taxonomy, and discuss its phylogenetic position in more detail.

## MATERIALS AND METHODS

The phylogenetic relationships of *Guchengosuchus shiguaiensis* were analyzed using the phylogenetic dataset of *Ezcurra (2016)* as modified by subsequent authors (*Ezcurra et al., 2017*; *Nesbitt et al., 2017*; *Sengupta, Ezcurra & Bandyopadhyay, 2017*; *Stocker et al., 2017*; *Ezcurra & Butler, 2018*; *Ezcurra et al., 2018*). This data matrix is composed of 116 active terminals and 694 active characters (character 119 was deactivated before the tree searches following *Ezcurra et al., 2017*). Here, an additional character state was added to characters 46 and 393 and a few scorings were changed for these characters and for characters 15, 56, 69, and 652 (see Appendix and Supplementary Material). The matrix was analyzed under equally weighted maximum parsimony using TNT v.1.5 (*Goloboff, Farris & Nixon, 2008*; *Goloboff & Catalano, 2016*). The search initially used a combination of tree-search algorithms including Wagner trees, TBR branch swapping, sectorial searches, Ratchet (perturbation phase stopped after 20 substitutions) and Tree Fusing (five rounds), until 100 hits of the same minimum tree length were achieved. The best trees obtained were subjected to a final round of TBR branch swapping. Zero length branches in any of the recovered most parsimonious trees (MPTs) were collapsed. The following characters were considered additive (ordered) during the searches, because they represent nested sets of homologies: 1, 2, 7, 10, 17, 19–21, 28, 29, 36, 40, 42, 46, 50, 54, 66, 71, 74–76, 122, 127, 146, 153, 156, 157, 171, 176, 177, 187, 202, 221, 227, 263, 266, 278, 279, 283, 324, 327, 331, 337, 345, 351, 352, 354, 361, 365, 370,

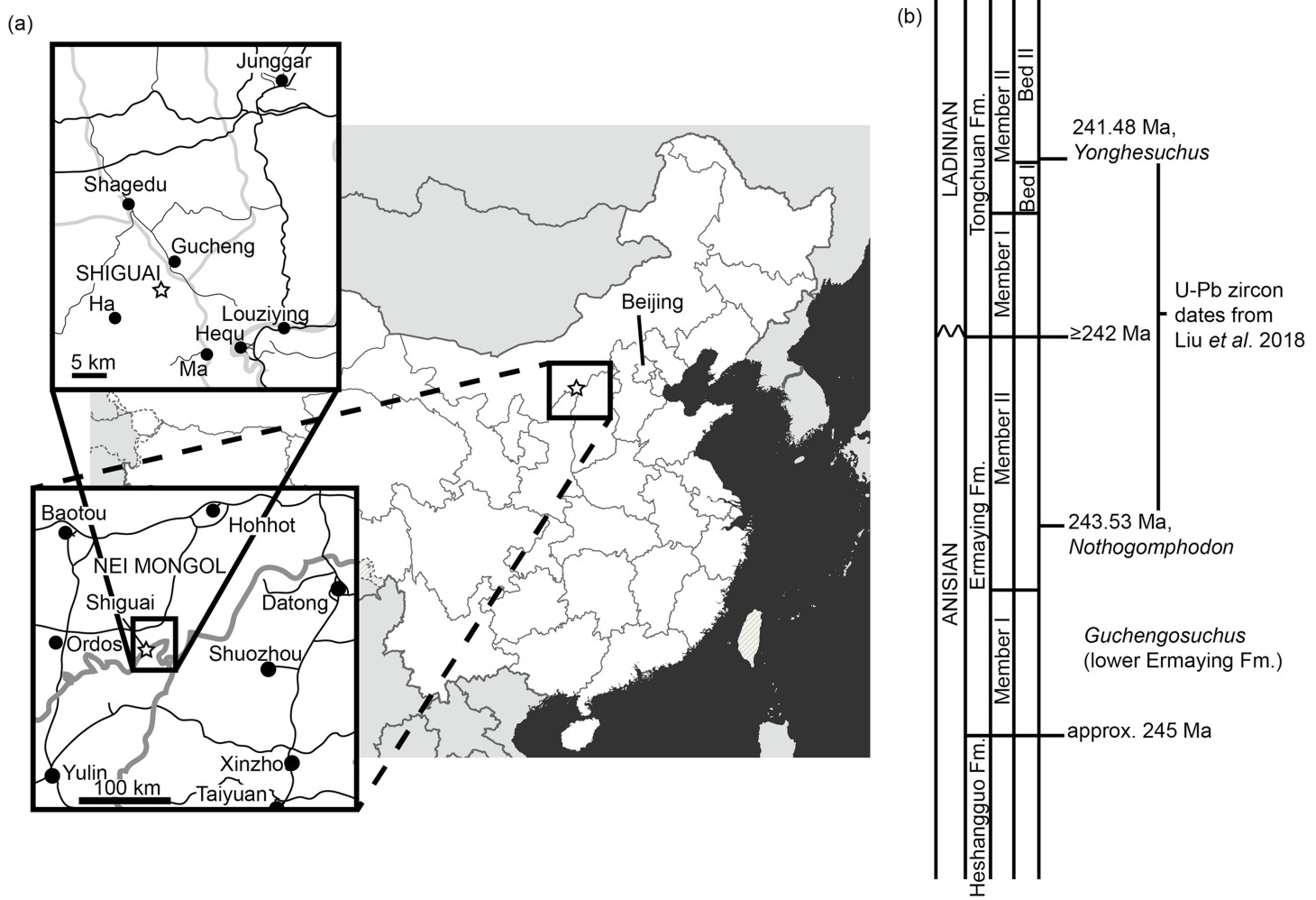

**Figure 1 Locality information for *Guchengosuchus shiguaiensis*.** (A) Maps showing the location of the locality of *Guchengosuchus*. The exact coordinates of the locality are unknown, but it is reported as being near the village of Shiguai in Gucheng township. Star indicates Shiguai on each map. Colors in largest-scale map: white, People's Republic of China; light gray, other countries; dark gray, ocean; thick gray lines, national borders; thin gray lines, province borders. Colors in larger-scale inset: black lines, roads; gray lines, province borders; black circles, major settlements. Colors in smaller-scale inset: larger black lines: larger roads; smaller black lines: smaller roads; gray lines, larger rivers; black circles, named settlements. (B) Stratigraphic column based on figure 1 of *Liu et al. (2018)* showing correspondence of the Ermaying Formation to international Middle Triassic stage dates. *Guchengosuchus* is reported to be from the lower part of the Ermaying Formation, but no more precise information is available.

377, 379, 386, 387, 398, 410, 424, 430, 435, 446, 448, 454, 458, 460, 463, 470, 472, 478, 482, 483, 485, 489, 490, 504, 510, 516, 529, 537, 546, 552, 556, 557, 567, 569, 571, 574, 581, 582, 588, 648, 652, and 662. Branch support was quantified using decay indices (Bremer support values) and a bootstrap resampling analysis, using 1,000 pseudoreplicates and reporting both absolute frequency and GC frequency (i.e., the difference between the frequencies of recovery in pseudoreplicates of the clade in question and the most frequently recovered contradictory clade) for each clade (*Goloboff et al., 2003*). The minimum number of additional steps necessary to generate alternative, suboptimal tree topologies was calculated by constraining the position of *Guchengosuchus shiguaiensis* in different parts of the tree and rerunning the analysis.

Stereopair photographs of some of the elements of the holotype skeleton of *Guchengosuchus shiguaiensis* are provided as Supplementary Material, as are reproductions of line drawings by *Peng (1991)* of parts of the skeleton that are currently unavailable for study. Michael Parrish provided photographs of some parts of the skeleton taken during a visit to the IVPP in 1990 that show some of the currently unavailable elements of the skeleton; these photographs are also reproduced in the Supplementary Material.

## SYSTEMATIC PALAEONTOLOGY

ARCHOSAURIFORMES *Gauthier, Kluge & Rowe, 1988*
ERYTHROSUCHIDAE *Watson, 1917* sensu *Ezcurra, Lecuona & Martinelli, 2010*
*Guchengosuchus Peng, 1991*

**Type species.** *Guchengosuchus shiguaiensis Peng, 1991*.

**Generic diagnosis.** As for type and only known species.

*Guchengosuchus shiguaiensis Peng, 1991*

**Holotype.** IVPP V8808: left maxilla, partial skull roof, left pterygoid, partial braincase, posterior portion of the right hemimandible, two anterior–middle cervical vertebrae; one probable anterior dorsal lacking most of the centrum; a fragment of presacral vertebra; four cervical and dorsal ribs; partial right scapula, humerus, ulna, and radius, a metapodial, and an ungual phalanx. Several of the bones originally figured and described by *Peng (1991)* as part of the holotype could not be located in the collection of the IVPP and may be lost (scapula and limb bones; see below), and other bones have been damaged since the original description (maxilla and pterygoid).

**Locality.** Shiguai Village, Gucheng Township, Fugu County, Shaanxi Province, People's Republic of China (*Peng, 1991*; Fig. 1). *Peng (1991)* reported that all the elements were collected from a 1 m$^2$ area and belong to a single individual.

**Stratigraphic horizon.** Lower part of the Ermaying Formation (*Peng, 1991*). Earliest Anisian (older than 243.53 Ma), early Middle Triassic (*Liu et al., 2018*; Fig. 1).

**Emended diagnosis.** Medium-sized archosauriform distinguished from other archosauromorphs by the following unique combination of features (autapomorphies indicated with an asterisk): maxilla with 14 tooth positions and ankylothecodont tooth implantation; maxilla without maxillo-nasal tuberosity and antorbital fossa; nasal with series of three knob-like convexities dorsal to the facet for the postnarial process of the premaxilla*; parietal with dorsal surface of the base of anterolateral process bearing a subtriangular fossa that extends onto the frontal and is separated from the supratemporal fenestra by a raised edge*; pterygoid without palatal teeth; anterior–middle cervical vertebrae with a strongly transversely convex and rugose distal expansion of the

neural spine*; accessory lamina subdivides the postzygapophyseal centrodiapophyseal fossae of the cervical and anterior dorsal vertebrae*; accessory articular surface for third head of cervical rib positioned at the same height dorsoventrally as the diapophysis*; and scapular blade with strongly concave posterior margin (modified from *Ezcurra, 2016*).

## ANATOMICAL DESCRIPTION

The currently available bones of IVPP V8808 are described and compared in detail based on our own observations, while bones that are currently unavailable or portions of bone that have been damaged since the original description are compared with those of other early archosauriforms based on the original description and figures of *Peng (1991)*.

**Maxilla.** The left maxilla is present (Figs. 2 and 3; see also Supplementary Material), and was originally nearly complete and well preserved (*Peng, 1991*, fig. 1, pl. 1.5; Fig. 4; Supplementary Material), but unfortunately the ascending process has mostly been lost since its original description, and the crowns of maxillary teeth 2, 4, and 6 have also been damaged. The bone is slightly incomplete both at its anterior margin and posteriorly, along the surface for contact with the jugal. As preserved, the maxilla has a length of 165 mm. The minimum dorsoventral height of the maxilla below the antorbital fenestra is 35 mm.

The main body of the maxilla is elongate and dorsoventrally narrow. Its ventral margin is very gently convex along the anterior process in lateral view, meaning that the anterior end of the tooth row is slightly upturned relative to the tooth row midpoint. The ventral margin of the maxilla along the horizontal process is straight, but this margin is concave in *Garjainia prima* (PIN 2394/5, 951/34), *Erythrosuchus africanus* (BP/1/4680, 5207), and *Chalishevia cothurnata* (PIN 4366/1). The lateral surface of the maxilla is flat. The anterior process of the maxilla is anteroposteriorly short, representing approximately 35% of the total length of the bone as preserved. The dorsal margin of the anterior process and the anterior margin of the ascending process are continuous along a curve that is slightly concave in lateral view (*Peng, 1991*, fig. 1, pl. 1.5), but to a lesser degree than in *Chalishevia cothurnata* (PIN 4366/1) and some specimens of *Shansisuchus shansisuchus* (e.g., IVPP V2508). By contrast, the anterior process is well distinguished from the ascending process by a clear inflexion in *Garjainia prima* (PIN 2394/5, 951/32), *Erythrosuchus africanus* (BP/1/2529, 5207), *Shansisuchus shansisuchus* (*Young, 1964*, figs. 10, 11), and *Chalishevia cothurnata* (PIN 4366/1). There is no indication of a facet for articulation with the premaxilla along this concave margin, which *Peng (1991)* identified as the edge of an accessory opening between the premaxilla and the maxilla ("secondary antorbital fenestra") (but see Discussion).

The base of the ascending process is situated above crowns 2–5. Because of the damage to the anterior end of the maxilla, it is not possible to determine if there was a short edentulous section adjacent to the premaxilla, or to confirm the presence or absence of an anteriorly opening notch and/or groove. The ascending process is anteroposteriorly broad and slightly thickened at its posterior margin (Fig. 3). This thickening curves posteroventrally at its base, following the margin of the antorbital fenestra, and fades out

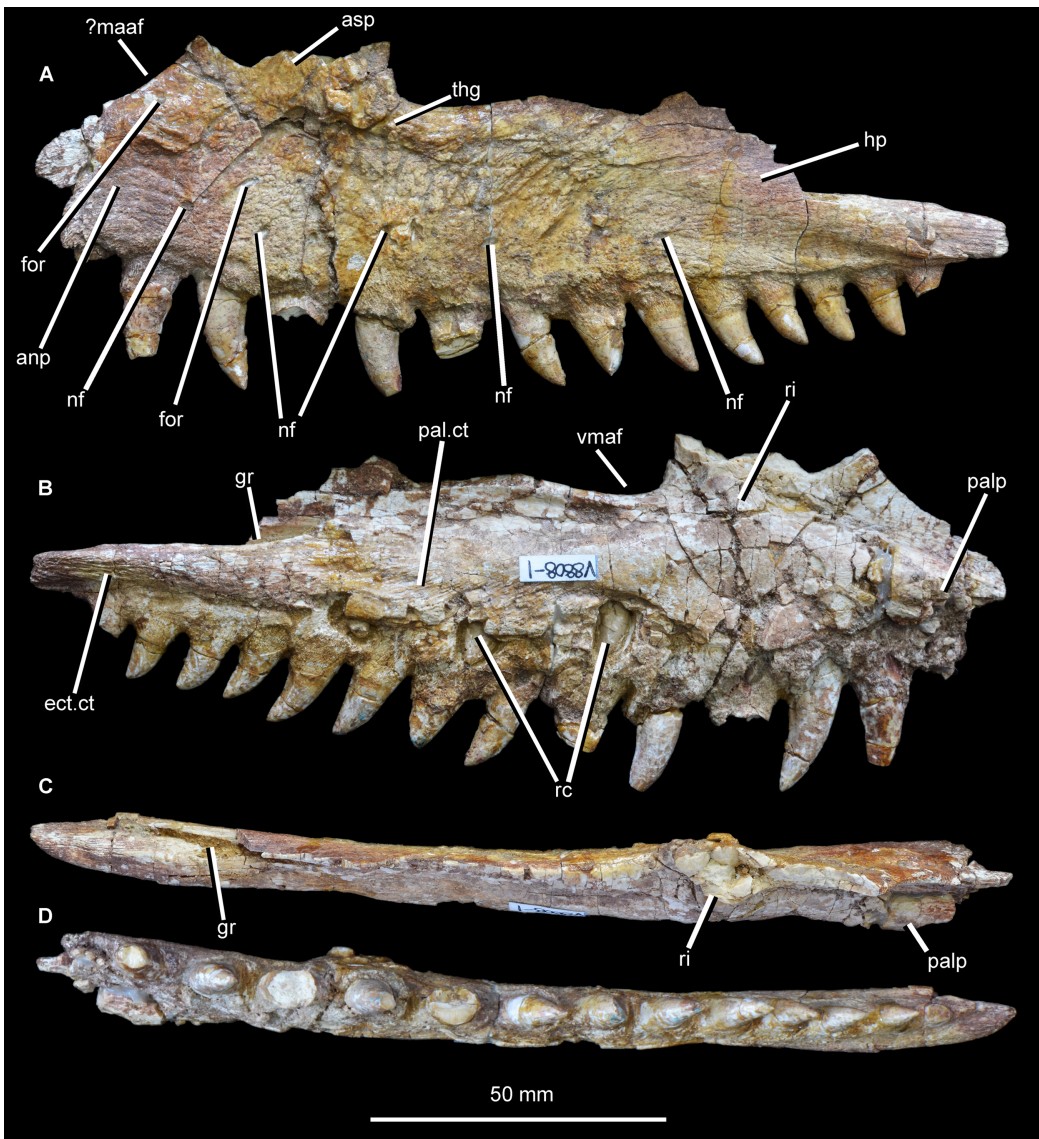

**Figure 2 Left maxilla of *Guchengosuchus shiguaiensis*, IVPP V8808, in lateral (A), medial (B), dorsal (C), and ventral (D) views.** Abbreviations: anp, anterior process; asp, ascending process; ect.ct, contact surface for the ectopterygoid; for, foramina; gr, groove; hp, horizontal process; ?maaf, proposed margin of the accessory antorbital fenestra identified by *Peng (1991)*—see Discussion for details; nf, nutrient foramina; palp, base of broken palatal process; pal.ct, contact surface for the palatine; rc, replacement crown; ri, ridge; thg, thickening of maxilla along the posterior margin of the ascending process and anteroventral margin of the antorbital fenestra; vmaf, ventral margin of the antorbital fenestra.

immediately ventral to the anterior end of the antorbital fenestra, similar to the condition in *Garjainia prima* (PIN 2394/5). The lateral thickening of the ascending process is considerably less developed than the probably homologous pillar-like maxillo-nasal tuberosity (sensu *Ezcurra, 2016*) of *Erythrosuchus africanus* (BP/1/2529, 5207), *Garjainia prima* (PIN 2394/5, 951/32), *Shansisuchus shansisuchus* (IVPP V2501, V2503), and *Chalishevia cothurnata* (PIN 4366/1). The preserved ventral margin of the antorbital fenestra extends for around 24 mm and is slightly concave dorsally, resembling the
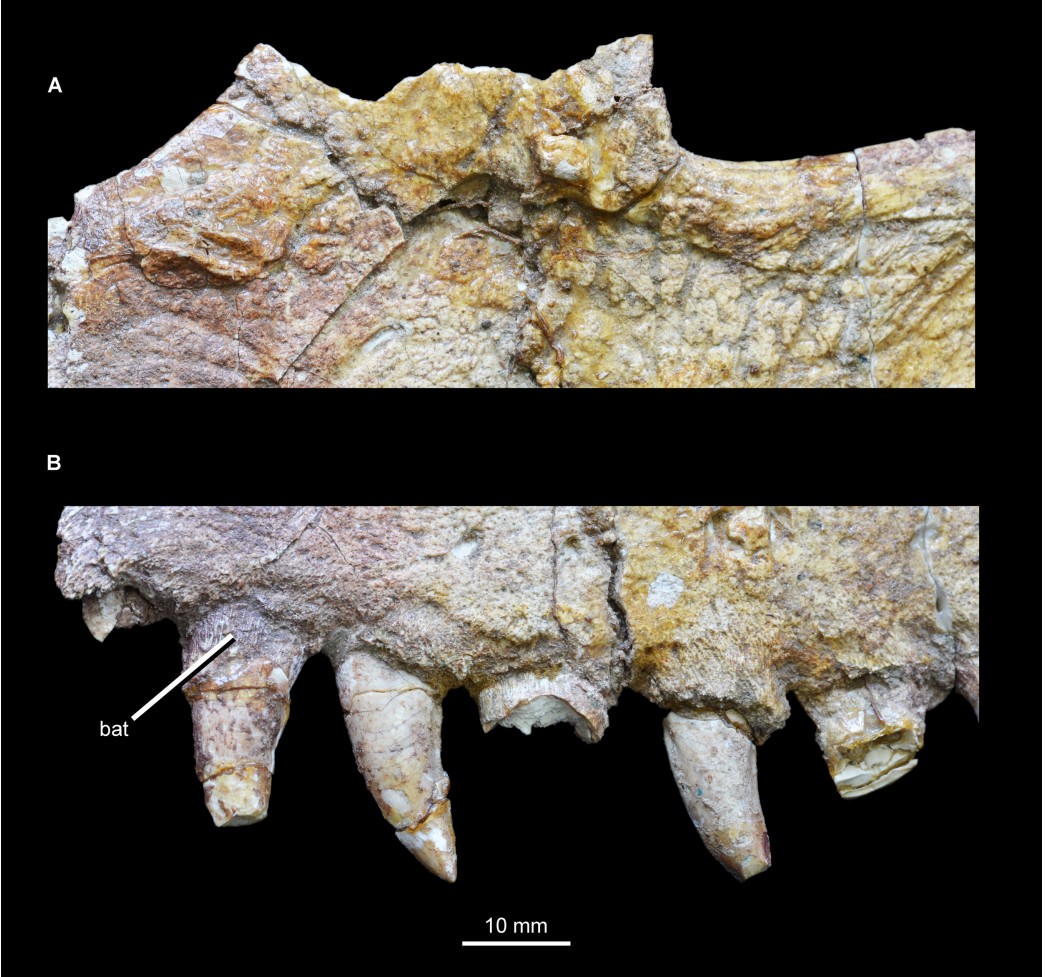

**Figure 3 Close-ups of the left maxilla of *Guchengosuchus shiguaiensis*, IVPP V8808, focusing on the base of the ascending process (A) and tooth positions 1–6 (B).** Abbreviation: bat, bone of attachment.

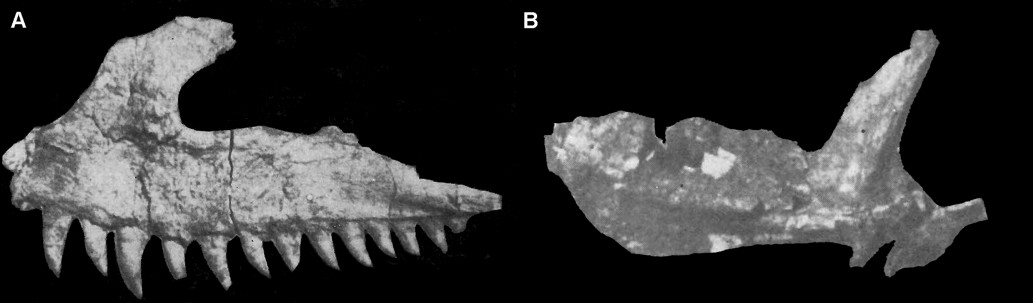

**Figure 4 Selected cranial elements of *Guchengosuchus shiguaiensis*, IVPP V8808, as originally preserved and figured by *Peng (1991)*.** Photographs show the left maxilla in lateral view (A) and the left pterygoid in ventral view (B). Both elements have been damaged since their description by *Peng (1991)*—see Figs. 2 and 10 for details. *Peng (1991)* did provide reduction factors (e.g., *x*½) for individual bones in his plates; however, the accuracies of these are unclear. As such, the present figure should not be used to estimate relative proportions of individual bones.

condition in several basal archosauriforms (e.g., *Proterosuchus fergusi*: BSPG 1934 VIII 514, RC 96, SAM-PK-11208; *Fugusuchus hejiapanensis*: *Cheng, 1980*, fig. 22; *Erythrosuchus africanus*: BP/1/5207). As a result, the horizontal process increases slightly in height posterior to the anterior border of the antorbital fenestra. Beyond this point the ventral margin of the antorbital fenestra is broken, so that the maxilla appears to generally taper in dorsoventral height toward its posterior end, but the true profile of the maxilla cannot be determined. At its posterior end, above crowns 11–14, the maxilla is slightly thickened below the area of articulation with the lacrimal and jugal, and this thickening extends from anterodorsal to posteroventral. No antorbital fossa is evident on either the main body of the maxilla or the posterior surface of the base of the ascending process, similar to the condition in *Proterosuchus fergusi* (BSPG 1934 VIII 514, RC 96, SAM-PK-11208), *Kalisuchus rewanensis* (NM QR 3570), and *Fugusuchus hejiapanensis* (*Cheng, 1980*, fig. 22). In *Chalishevia cothurnata* (PIN 4366/1) and *Shansisuchus shansisuchus* (IVPP V2501, V2503), by contrast, a well-developed antorbital fossa is present on the base of the ascending process and horizontal process. An antorbital fossa is also present in *Erythrosuchus africanus* (BP/1/5207), but is less extensive.

A number of circular foramina pierce the lateral surface of the maxilla. A row of small, irregularly spaced nutrient foramina, seven of which are clearly identifiable although it seems possible that nine or more were present originally, occurs approximately 10 mm above the alveoli. These foramina are positioned in a single row aligned roughly parallel to the tooth row. The anteriormost foramen is positioned above crown 2 and the posteriormost above crown 9, the latter opening mainly posteriorly. The other foramina open laterally and ventrally, and in several cases give rise to short grooves that extend ventrally toward the tooth row, resembling the condition in *Erythrosuchus africanus*, *Garjainia prima*, and *Chalishevia cothurnata* (*Ezcurra, 2016*). An additional small foramen is positioned immediately below the concave anterodorsal margin of the anterior process, and yet another lies dorsal to the main row of foramina and below the ascending process at the anteroposterior level of alveolus 3, opening anteroventrally. The texture of the lateral surface of the maxilla is notably rugose, and the rugosity is best developed below the antorbital fenestra and the posterior edge of the base of the ascending process. This rugose texture includes a number of dorsally or posterodorsally trending grooves that emanate from the row of nutrient foramina.

The majority of the medial surface of the main body of the maxilla is dorsoventrally convex, resulting in transverse thickening of the maxilla. A thickened ridge extends dorsally from the main body onto the posterior half of the medial surface of the ascending process. Anterior to this thickened ridge, the ascending process forms a transversely compressed and laterally offset sheet, resembling the condition in other archosauriforms (e.g., *Kalisuchus rewanensis*: QM F8998). Most of this anterior sheet is currently missing, but as figured by *Peng (1991*, fig. 1B*)* its medial surface appears to have lacked the deep, well-rimmed fossa present in several eucrocopodan archosauriforms (e.g., *Euparkeria capensis*: SAM-PK-K6050; *Gow, 1970*; *Teleocrater rhadinus*: *Nesbitt et al., 2017*; *Asilisaurus kongwe*: *Nesbitt et al., 2017*; *Silesaurus opolensis*: ZPAL Ab III/361/26). A similar, transversely compressed and laterally offset sheet is present posterior to the

ascending process, forming the dorsal part of the main body of the maxilla, and bounding the antorbital fenestra ventrally. There are no foramina visible on the medial surface, and the palatal process is broken at its base. The preserved portion of the palatal process is situated immediately above the bases of the interdental plates, resembling the condition in several other archosauriforms (e.g., *Proterosuchus goweri*: NM QR 880; "*Chasmatosaurus*" *yuani*: IVPP V36315; *Kalisuchus rewanensis*: QM F8998; *Garjainia prima*: PIN 2394/5; *Euparkeria capensis*: SAM-PK-6050). By contrast, this process is placed distinctly dorsal to the alveolar margin and adjacent to the anterodorsal margin of the bone in *Erythrosuchus africanus* (BP/1/4680, SAM-PK-K1098), *Asperoris mnyama* (NHMUK PV R36615), and several archosaurs (e.g., *Yarasuchus deccanensis*: ISIR 334-2; *Teleocrater rhadinus*: *Nesbitt et al., 2017*; *Batrachotomus kupferzellensis*: SMNS 52970; *Herrerasaurus ischigualastensis*: PVSJ 53). The posterior end of the horizontal process is considerably thickened, and possesses a strongly dorsoventrally convex medial surface. A cluster of anterodorsally-to-posteroventrally oriented thin ridges occurs immediately above the bases of the interdental plates, at the level of the sixth to ninth alveoli. This striated surface represents the facet for reception of the posterolateral process of the palatine. The posterior tip of the horizontal process is also covered with a series of thin ridges. In this case, however, the ridges are longitudinal and mark the area of contact with the lateral process of the ectopterygoid, as also occurs in various other saurian reptiles (e.g., *Gephyrosaurus bridensis*: *Evans, 1980*; *Trilophosaurus buettneri*: *Spielmann et al., 2008*; *Rhynchosaurus articeps*: NHMUK PV R1236; *Garjainia prima*: PIN 2394/5; *Chanaresuchus bonapartei*: PULR 07; *Doswellia kaltenbachi*: USNM 214823; *Parasuchus angustifrons*: BSPG 1931 X 502). Dorsal to the thickening of the posterior end of the horizontal process there is a very deep, longitudinal groove that received the jugal and possibly also the lacrimal if the jugal did not participate in the border of the antorbital fenestra. The position of this facet suggests that the antorbital fenestra was relatively long anteroposteriorly.

A total of 14 teeth were present (assuming that the most anterior preserved crown was the most anterior in the complete specimen), a relatively low maxillary tooth count comparable to those for *Garjainia prima* (PIN 2394/5: maxillary tooth count 14 or possibly 13), *Erythrosuchus africanus* (BP/1/5207: tooth count 11), *Chalishevia cothurnata* (PIN 4366/1; tooth count 12 or possibly 13), *Shansisuchus shansisuchus* (*Young (1964)* described 9 or possibly 10 tooth positions and *Wang et al. (2013)* described probably 13 teeth), and *Euparkeria capensis* (*Ewer, 1965*; tooth count 13). By contrast, higher maxillary tooth counts are present in *Tasmaniosaurus triassicus* (UTGD 54655; >21), *Proterosuchus fergusi* (BP/1/3993, BSPG 1934 VIII 514, GHG 231 RC 59, 96, SAM-PK-11208, K140, K10603; tooth count 20–31, depending on ontogenetic stage, *Ezcurra & Butler (2015)*), "*Chasmatosaurus*" *yuani* (>23 in IVPP V90002 and ≥29 in IVPP V2719) and *Prolacerta broomi* (BP/1/471, *Modesto & Sues, 2004*; tooth count 24–25). The largest teeth are situated immediately anterior to the level of the anterior border of the antorbital fenestra, in tooth positions 2–4. The teeth are all (with the possible exception of crown 3) fused to their alveoli by bone of attachment (ankylothecodont tooth

implantation), which is covered with fine apicobasally extending striations (Fig. 3). However, it cannot be determined whether the teeth are set in deep sockets, resembling the condition in allokotosaurians (e.g., *Azendohsaurus madagaskarensis*: *Nesbitt et al., 2015*; *Shringasaurus indicus*: *Sengupta, Ezcurra & Bandyopadhyay, 2017*), *Prolacerta broomi* (*Modesto & Sues, 2004*), proterosuchids (e.g., *Proterosuchus fergusi*: BSPG 1934 VIII 514, RC 59, TM 201), *Kalisuchus rewanensis* (QM F8998), *Garjainia madiba* (BP/1/5525) and referred specimens of *Garjainia prima* (PIN 951/32, 34, 55). By contrast, maxillary tooth implantation appears to be fully thecodont (i.e., deep alveoli and absence of ankylosis) in the holotype of *Garjainia prima* (PIN 2394/5), *Erythrosuchus africanus* (BP/1/2529, 4680), and eucrocopodan archosauriforms (*Ezcurra, 2016*), although computed tomographic data would be useful to more fully examine implantation in the former two species. Several replacement crowns are visible on the medial surface, above teeth 4, 6, 8, and 10, suggesting an alternating sequence of replacement. The maxillary tooth crowns resemble those of other carnivorous archosauriforms (e.g., *Garjainia prima*: PIN 2394/5; *Erythrosuchus africanus*: NHMUK PV R3592). Crowns are recurved, with their apices positioned distal to the distal margins of their bases, and labiolingually compressed. Fine mesial denticles are present, but are poorly preserved in most cases and appear to be restricted to the apical 30–40% of the crown. The distal denticles are generally better preserved, and appear to extend along almost the entire distal margin. The denticles have a rectangular outline in labial or lingual view. Enamel wrinkles and blood grooves are absent.

**Nasal.** An incomplete left nasal is preserved (Fig. 5; Supplementary Material), and is 133 mm long anteroposteriorly as preserved. The nasal is missing the anterior tip, a substantial posterior portion including the contacts with the frontal and probably also the prefrontal, and parts of the lateral margin, but does not appear to have sustained any damage since its original description by *Peng (1991)*. The nasal is an anteroposteriorly elongated and transversely narrow bone with a straight, slightly dorsoventrally thickened medial margin bearing a groove for articulation with the opposite element. The external surface of the nasal can be divided into distinct medial and lateral parts with different orientations. The medial division is a largely dorsally facing surface that formed much of the anterior part of the skull roof. This surface has a gentle transverse convexity along most of its length, but at its posterior end becomes flattened to slightly transversely concave. The skull roof surface is notably rugose along most of its length, resembling the condition in *Garjainia prima* (PIN 2394/5), but being considerably less strongly sculptured than in *Asperoris mnyama* (*Nesbitt, Butler & Gower, 2013*). The dorsally exposed part of the nasal is broadest posteriorly (23 mm wide) and tapers gradually toward the anterior end of the element (15 mm wide above the facet for the postnarial process of the premaxilla). At the anterior end a narrow groove is present on the dorsal surface of the nasal adjacent to the midline. This feature may represent the articular facet for the prenarial process of the premaxilla, as occurs in other basal archosauriforms (e.g., "*Chasmatosaurus*" *yuani*: IVPP V36315; *Erythrosuchus goweri*: *Gower, 2003*; *Rhadinosuchus gracilis*: *Ezcurra, Desojo & Rauhut, 2014*).

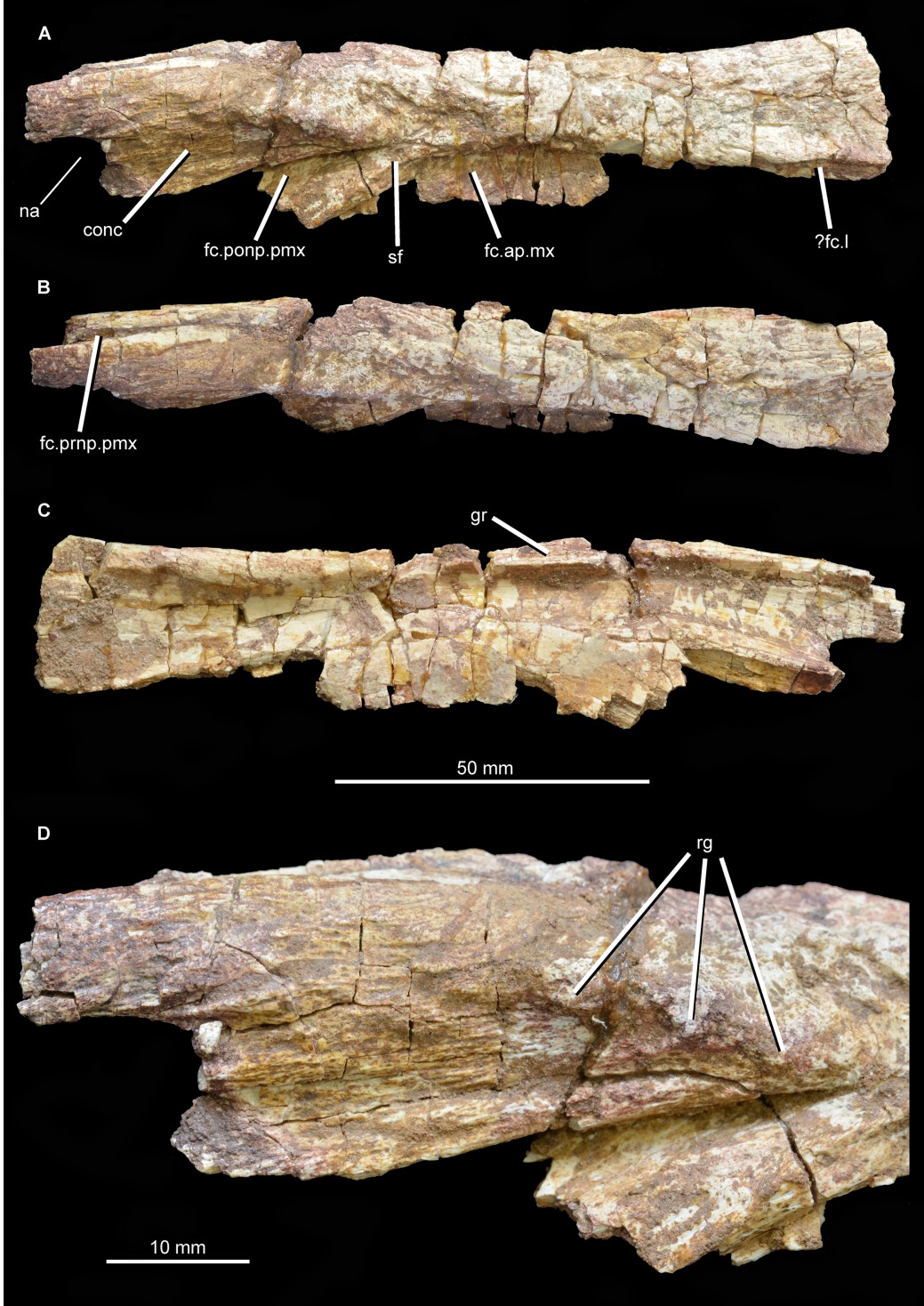

**Figure 5** **Left nasal of *Guchengosuchus shiguaiensis*, IVPP V8808, in lateral (A), dorsal (B), and medial (C) views, and with close-up of the anterior end in lateral (D) view.** Abbreviations: conc, concavity; fc.ap. mx, facet for the ascending process of the maxilla; fc.l, facet, possibly for articulation with the lacrimal; fc. ponp.pmx, facet for the postnarial process of the premaxilla; fc.prnp.pmx, facet for the prenarial process of the premaxilla; gr, groove on medial margin for articulation with opposing nasal; na, border of external naris; rg, knob-like rugosities; sf, surface separating the postnarial process of the premaxilla from the ascending process of the maxilla.

The lateral part of the external surface of the nasal forms the dorsal part of the lateral wall of the preorbital part of the skull, and thus would have faced mostly laterally and slightly dorsally. At the anterior end a small part of the posterodorsal margin of the external naris is preserved. Posteroventral to this, the external surface of the nasal between the external naris and the facet for the postnarial process of the premaxilla is gently concave. This concavity runs parallel to the facet for the postnarial process and is terminated posterodorsally by a short anteroposteriorly extending series of three knob-like rugosities (Fig. 4D) that are not present in *Chalishevia cothurnata* (PIN 4366/1), *Shansisuchus shansisuchus* (*Young, 1964*; *Wang et al., 2013*), *Garjainia prima* (PIN 2394/5), or *Erythrosuchus africanus* (BP/1/5207; NM QR 1473; NHMUK PV R3592). Accordingly, this condition appears to be an autapomorphy of *Guchengosuchus*.

Posteroventral to this concavity is the posterodorsally tapering facet for the postnarial process of the premaxilla. This facet is deepest anterodorsally, and becomes shallower posteroventrally. The facet is relatively narrow, suggesting that the postnarial process was also relatively narrow at its tip as in *Chalishevia cothurnata* (PIN 4366/1), *Shansisuchus shansisuchus* (*Wang et al., 2013*), and *Erythrosuchus africanus* (BP/1/5207). By contrast, in *Garjainia prima* the distal tip of the postnarial process of the premaxilla is comparatively broad (PIN 2394/5). Posteroventrally, a very narrow, almost flat surface, representing the lateral exposure of the descending process of the nasal, separates the facet for reception of the postnarial process of the premaxilla from the facet for reception of the ascending process of the maxilla (see *Ezcurra, 2016*, character 81). A similarly narrow descending process is present in *Erythrosuchus africanus* (BP/1/5207) and several other archosauriforms (e.g., *Asperoris mnyama*: NHMUK PV R36615; *Euparkeria capensis*: SAM-PK-5867; *Turfanosuchus debanensis*: IVPP V3237; *Gracilisuchus stipanicicorum*: MCZ 4117). By contrast, the descending process of the nasal is anteroposteriorly very broad in *Garjainia prima* (PIN 2394/5), *Shansisuchus shansisuchus* (*Young, 1964*; *Wang et al., 2013*), and *Chalishevia cothurnata* (PIN 2867/7).

Posterior to the descending process of the nasal there is a diagonal ridge, anteroventrally oriented, which delimits anteriorly the facet for reception of the ascending process of the maxilla. This facet becomes deeper posteriorly. As a result, the facet for reception of the ascending process of the maxilla is a medially inset, transversely compressed, ventrally descending sheet of bone situated slightly posterior to the facet for the postnarial process of the premaxilla. This surface for the maxilla is broken posteriorly. The posteriormost preserved part of the lateral margin of the nasal is grooved, perhaps for contact with the lacrimal. The internal surface of the nasal is transversely concave, with no notable ridges or foramina present.

**Frontal.** The frontals are largely intact and preserved as part of the articulated section of skull roof, but their anterior ends are missing (Figs. 6 and 7; see also Supplementary Material). The pair of frontals is considerably anteroposteriorly longer than wide, as occurs in *Prolacerta broomi* (BP/1/471), *Teyujagua paradoxa* (*Pinheiro et al., 2016*), *Proterosuchus fergusi* (RC 59, 96, BP/1/3993, SAM-PK-K10603), *Tasmaniosaurus triassicus* (*Ezcurra, 2014*; UTGD 54655), *Fugusuchus hejiapanensis*

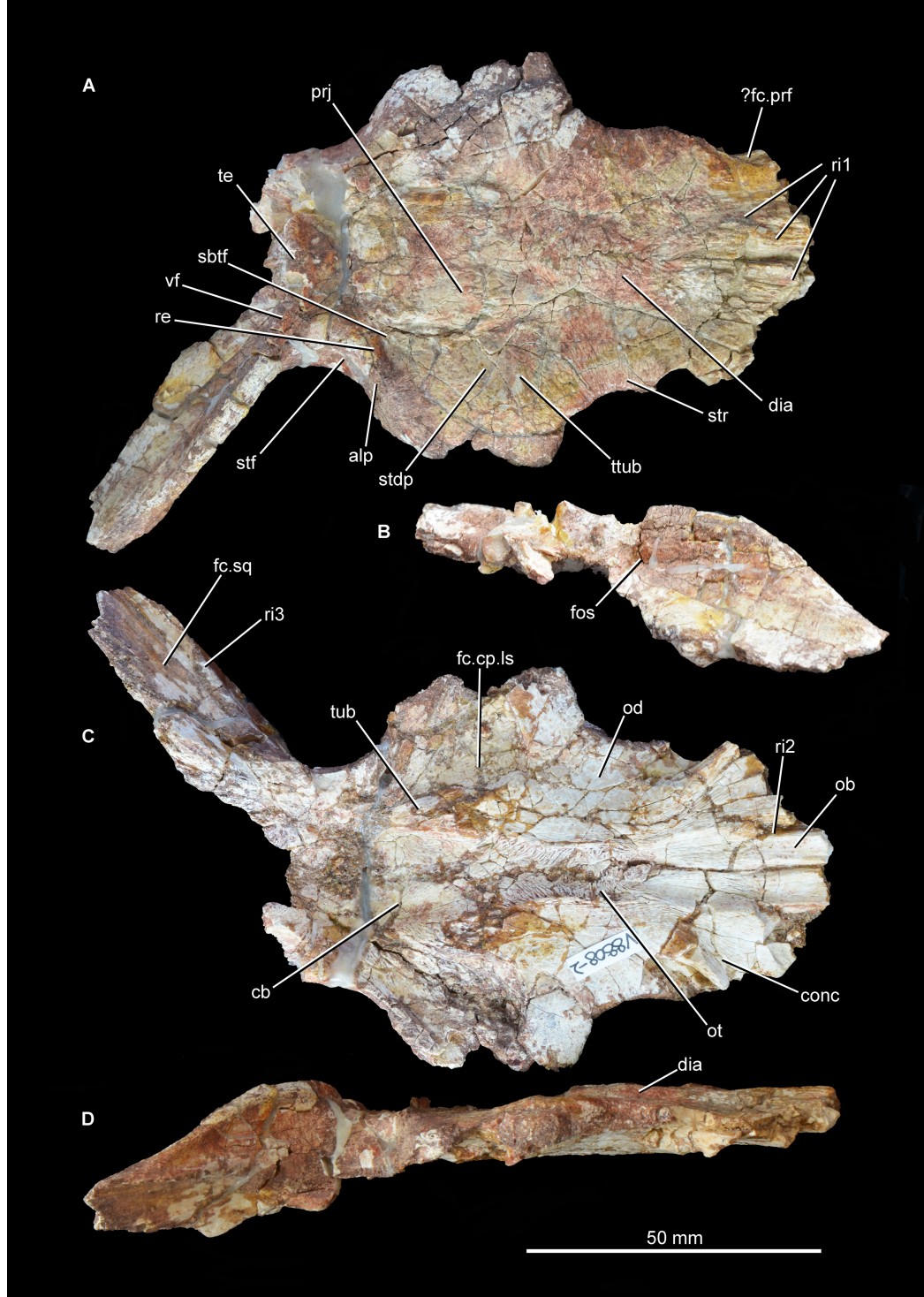

**Figure 6 Skull roof of *Guchengosuchus shiguaiensis*, IVPP V8808, in dorsal (A), posterior (B), ventral (C), and right lateral (D) views.** Abbreviations: alp, anterolateral process of the parietal; cb, impression of the cerebrum; conc, concavity between the impression of the olfactory bulbs and the orbital depression; dia, dorsally inflated area of the nasals; fc.cp.ls, facet for the capitate process of the laterosphenoid; ?fc.prf, possible facet for prefrontal; fc.sq, facet for the squamosal; fos, fossa on the base of the posterolateral process of the parietal; ob, impression of olfactory bulbs; od, orbital depression; ot, olfactory tract; prj, median projection of the parietals; re, raised edge on parietal posteriorly delimiting a

**Figure 6** (continued)
subtriangular fossa; ri1, longitudinal ridges on the anterior parts of the frontals; ri2, ridge laterally delimiting impression of the olfactory bulbs; ri3, ridge on the anterior surface of the posterolateral process of the parietal; sbtf, subtriangular fossa on parietal; stdp, subtriangular depression on posterolateral corner of frontal; stf, supratemporal fossa; str, striations; te, transverse eminence on posterior margin of parietals; ttub, transverse tuberosity on the frontal; tub, tuberosity separating impression of cerebrum from the facet for the capitate process of the laterosphenoid; vf, vertical flange.

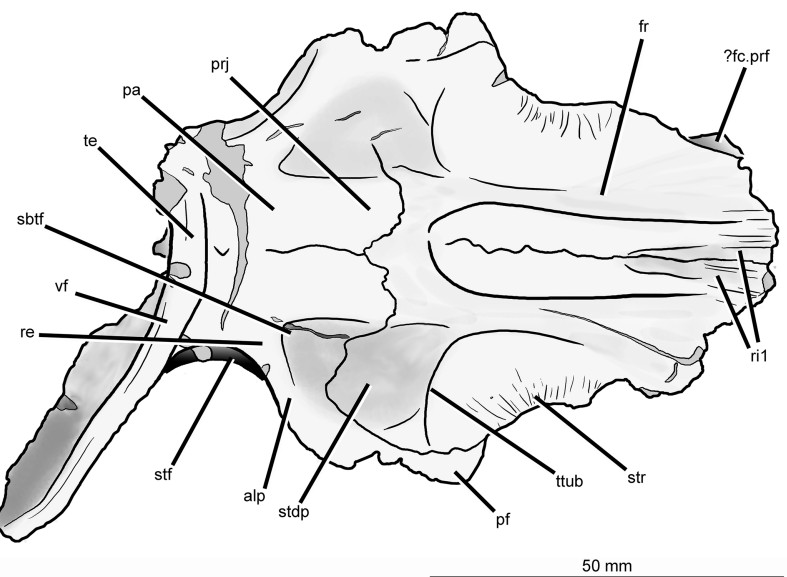

**Figure 7 Skull roof of *Guchengosuchus shiguaiensis*, IVPP V8808, in dorsal view.** Abbreviations: alp, anterolateral process of the parietal; dia, dorsally inflated area of the nasals; ?fc.prf, possible facet for prefrontal; fos, fossa on the base of the posterolateral process of the parietal; fr, frontal; pa, parietal; pf, postfrontal; prj, median projection of the parietals; re, raised edge on parietal posteriorly delimiting a subtriangular fossa; ri1, longitudinal ridges on the anterior parts of the frontals; sbtf, subtriangular fossa on parietal; stdp, subtriangular depression on posterolateral corner of frontal; stf, supratemporal fossa; str, striations; te, transverse eminence on posterior margin of parietals; ttub, transverse tuberosity on the frontal; vf, vertical flange.

(*Cheng, 1980*: fig. 22), *Erythrosuchus africanus* (*Gower, 2003*; NHMUK PV R3592, NM QR 1473), *Shansisuchus shansisuchus* (*Young, 1964*, figs. 1–6), and *Euparkeria capensis* (SAM-PK-5867). By contrast, the paired frontals are wider than long in *Garjainia prima* (PIN 2394/5). The dorsal surface of the most anteriorly preserved region of the frontal bears a few low longitudinal ridges. The facets for articulation with the prefrontals are not well preserved but may be situated at the anterolateral corners of the preserved portion of the frontals. The absence of a facet on the lateral surface of the frontal indicates that the bone contributed extensively to the dorsal border of the orbit, resembling the condition in most early archosauromorphs (e.g., *Prolacerta broomi*: BP/1/471; *Proterosuchus fergusi*: RC 96, BP/1/3993, SAM-PK-K10603; "*Chasmatosaurus*" *yuani*: IVPP V4067; *Sarmatosuchus otschevi*: *Gower & Sennikov, 1997*; *Euparkeria capensis* (SAM-PK-5867). By contrast, in other erythrosuchids the contribution of the frontal to the dorsal border of the orbit is very restricted or absent (*Fugusuchus hejiapanensis*:

*Cheng, 1980*; *Garjainia prima*: PIN 2394/5; *Erythrosuchus africanus*: *Gower, 2003*, NHMUK PV R3592, NM QR 1473; *Shansisuchus shansisuchus*: IVPP V2504, *Young, 1964*, fig. 5, *Wang et al., 2013*, fig. 2). The orbital margin of the frontal becomes dorsoventrally thicker posteriorly, toward the contact with the postfrontal. The dorsal surface of the frontal adjacent to the orbital border is slightly dorsally inflated and has several very thin and low striations oriented perpendicular to the margin of the bone. A second dorsally inflated area is situated on the midline of the skull roof on the posterior half of the paired frontals. The inflated area ends posteriorly at a point well anterior to the suture between the frontals and parietals, and bears a rugose ornamentation. This inflated area closely resembles in morphology and position an equivalent feature in *Garjainia prima* (PIN 2394/5), but is flanked laterally by a pair of moderately deep depressions. By contrast, no inflated median area is present in *Prolacerta broomi* (*Modesto & Sues, 2004*), *Proterosuchus fergusi* (RC 59, 96, BP/1/3993, SAM-PK-K10603), "*Chasmatosaurus*" *yuani* (IVPP V4067), *Fugusuchus hejiapanensis* (*Cheng, 1980*), *Erythrosuchus africanus* (*Gower, 2003*, NHMUK PV R3592, NM QR 1473), *Shansisuchus shansisuchus* (IVPP V2504, V2508), or *Euparkeria capensis* (SAM-PK-5867). In *Guchengosuchus shiguaensis* the inflated area terminates anteriorly at a median depression. The inflated areas on the midline and adjacent to the orbital border define a shallow concavity on the frontal that becomes slightly deeper anteriorly.

The dorsal surface of the posterolateral corner of the frontal possesses a subtriangular depression that extends posteriorly onto the parietal and is well delimited anteriorly by a transverse tuberosity. Posteriorly, the paired frontals do not enclose a pineal fossa, contrasting with the condition in *Garjainia prima* (PIN 2394/5) and some specimens of *Erythrosuchus africanus* (BP/1/5207, NM QR 1473, NHMUK PV R3592), and they meet the parietals along a suture that is poorly interdigitated medially. The parietals form a median, short subrectangular projection that extends anteriorly between the frontals. The lateral portion of the frontoparietal suture is strongly convex posteriorly, creating a posterior projection of the frontal resembling that present in *Shansisuchus shansisuchus* (IVPP V2508). The posterolateral corner of the frontal forms a laterally convex suture with the parietal and postfrontal, excluding contact between the frontal and the postorbital.

The ventral surface of the frontal displays a large and slightly concave orbital depression, which forms most of the roof of the orbit. This depression extends posteriorly onto the postfrontal. The olfactory tract extends anteroposteriorly between the orbital depressions, from which it is separated by distinct ridges (the crista cranii), and is hourglass-shaped. The olfactory tract opens anteriorly into a recess for the olfactory bulbs of the anterior brain, which is considerably anteroposteriorly longer than wide and laterally delimited by a ridge, as also occurs in other early archosauromorphs (e.g., *Shringasaurus indicus*: ISIR 781, 789, 790; *Prolacerta broomi*: BP/1/2675; *Tasmaniosaurus triassicus*: *Ezcurra, 2014*; *Sarmatosuchus otschevi*: PIN 2865/68). The latter ridge and the ridge that defines the anterior limit of the orbital depression bound a concave area at the anterolateral corner of the ventral surface of the frontal that is also present in some other basal archosauriforms (e.g., *Tasmaniosaurus triassicus*:

*Ezcurra, 2014*; *Sarmatosuchus otschevi*: PIN 2865/68). This condition differs from that in other erythrosuchids (e.g., *Garjainia prima*: *Huene, 1960*; *Erythrosuchus africanus*: *Gower, 2003*, NHMUK PV R3592, NM QR 1473; *Shansisuchus shansisuchus*: *Young, 1964*), in which there is a median longitudinal canal for the passage of the olfactory tract but no olfactory bulb impression bounded by a distinct, semilunate, posteromedially-to-anterolaterally oriented ridge. On the ventral surface of the skull roof, the suture between the frontal and the postfrontal is clearly visible on the left side but the frontoparietal suture is only partially discernible, being evident within the left facet for reception of the capitate process of the laterosphenoid and near the mid-line at the posterior end of the olfactory tract. The frontal forms most of the surface of the facet for reception of the capitate process. This facet is anteromedially very well delimited by a deep shelf that also forms the posterior border of the orbital depression, and anterolaterally defined by the posterior margin of the postfrontal.

**Postfrontal.** Both postfrontals are preserved in articulation with the frontal and parietal (Figs. 6 and 7; see also Supplementary Material), but they are lacking their ventral processes. The suture between the postfrontal and frontal is mainly anteroposteriorly oriented and slightly medially concave in dorsal view, closely resembling the condition in *Garjainia prima* (PIN 2394/5), *Shansisuchus shansisuchus* (IVPP V2508), and *Erythrosuchus africanus* (NHMUK PV R3592). This suture is V-shaped on the ventral surface of the skull roof of *Guchengosuchus shiguaiensis*. The postfrontal forms the posterodorsal corner of the orbit, and the ventral surface of the bone forms the posterior end of the orbital depression and is therefore slightly concave. The dorsal surface of the bone is slightly convex, and ornamented with low rugosities.

**Parietal.** The parietal possesses an anterolaterally projecting process that forms the anterior border of the supratemporal fenestra and articulates with the frontal and postfrontal (Figs. 6 and 7; see also Supplementary Material). The lateral end of this process is damaged. The dorsal surface of the base of the anterolateral process is occupied by a subtriangular fossa that extends onto the frontal. This feature does not represent an extension of the supratemporal fossa, being well separated from the border of the supratemporal fenestra by a raised edge adjacent to this opening. The medial border of the fossa is anteroposteriorly oriented, but poorly defined. To our knowledge, no similar fossa is present in other early archosauriforms. The dorsal surface of the parietals lacks a pineal fossa, as in some specimens of *Proterosuchus fergusi* (RC 59, BP/1/4016, 4224; *Ezcurra & Butler, 2015*) and in *Euparkeria capensis* (*Ewer, 1965*). By contrast, a pineal fossa is present in the skull roof of *Erythrosuchus africanus* (BP/1/5207, NM QR 1473, NHMUK PV R3592), *Garjainia prima* (PIN 2394/5), *Shansisuchus shansisuchus* (*Young, 1964*; IVPP V2501, V2504, V2508), *Proterosuchus alexanderi* (NM QR 1484), *Proterosuchus goweri* (NM QR 880), and some specimens of *Proterosuchus fergusi* (BP/1/3993, SAM-PK-K9957, SAM-PK-K10603, RC 96, TM 201, GHG 231). The supratemporal fossa is not dorsally exposed, being restricted to the lateral and anterolateral borders of the supratemporal fenestra, but is dorsoventrally very deep. The posterior margin of the

paired parietals forms a thick transverse eminence, as also occurs in other archosauromorphs (e.g., *Prolacerta broomi*, *Proterosuchus fergusi*, *Garjainia prima*) (*Müller, 2004*; *Ezcurra, 2016*). The posterolateral process of the parietal is posterolaterally directed at an angle of about 50° to the midline of the skull. The posterior eminence of the parietal continues onto the dorsal surface of the posterolateral process of the bone as a moderately low vertical flange, resembling the condition in *Prolacerta broomi* (BP/1/471), *Teyujagua paradoxa* (UNIPAMPA 653 cast), *Proterosuchus fergusi* (SAM-PK-K10603), *Fugusuchus hejiapanensis* (GMB V 313 photographs), *Shansisuchus shansisuchus* (*Young, 1964*, fig. 6), and *Euparkeria capensis* (SAM-PK-5867). By contrast, the posterolateral process acquires a wing-like appearance in occipital view with a strongly convex dorsal margin in *Garjainia prima* (PIN 2394/5) and *Erythrosuchus africanus* (*Gower, 2003*, NM QR 1473, NHMUK PV R3592). The posterior surface of the posterolateral process has a dorsoventrally concave curvature that deepens medially, terminating in a moderately deep, posteriorly opening fossa at the base of the process. It is not possible to determine whether the posterior surface of the process bears a tuberosity as in *Erythrosuchus africanus* (*Gower, 2003*). The anterior surface of the posterolateral process possesses a longitudinal, thick ridge. Above the ridge, the surface of the bone is slightly dorsoventrally concave, and below it there is a gently concave surface that represents the ventrally facing facet for the medial process of the squamosal. The facet for the squamosal possesses a number of low, poorly developed longitudinal ridges.

The ventral surface of the parietals possesses a deep concavity on the midline of the skull roof, representing the impression of the cerebrum. The concavity is laterally delimited by a thick tuberosity that separates it from the facet for reception of the capitate process of the laterosphenoid, as occurs in other archosauriforms (e.g., *Tasmaniosaurus triassicus*: *Ezcurra, 2014*; *Erythrosuchus africanus*: *Gower, 2003*; NM QR 1473). The ventral surface of the anterolateral process of the parietal forms part of the posterior end of the facet for reception of the laterosphenoid. The ventral surface of the posterolateral process is rugose, and articulated with the dorsolateral margin of the supraoccipital and the base of the paroccipital process. Because the left posterolateral process of the parietal is missing and the ventral margin of the distal half of the right process is broken, it is not possible to determine whether a post-temporal opening was originally present.

**Interparietal.** It is not possible to discern a suture between the parietals and the interparietal, assuming that the latter bone is present. A thickened region on the posterior surface of the skull roof might represent an interparietal, as occurs in other basal archosauriforms (e.g., *Proterosuchus alexanderi*: *Cruickshank, 1972*; *Erythrosuchus africanus*: *Gower, 2003*).

**Supraoccipital.** The supraoccipital is preserved in articulation with the otoccipital and the prootic (Fig. 8; see also Supplementary Material). A sharp median vertical ridge extends ventrally across the occipital surface of the supraoccipital from the posterior border of the skull roof, gradually becoming lower in the ventral direction and eventually

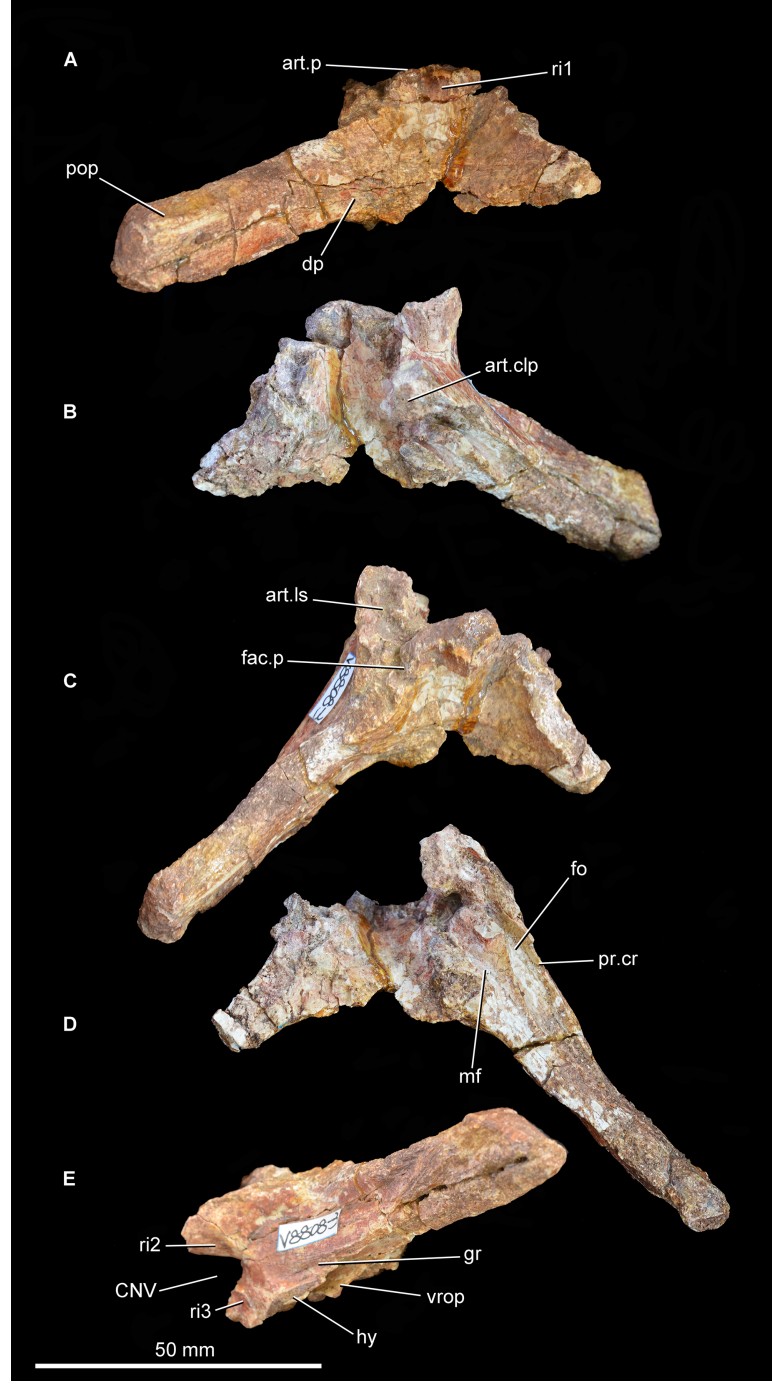

**Figure 8 Partial braincase of Guchengosuchus shiguaiensis, IVPP V8808, in posterior (A), anterior (B), dorsal (C), ventral (D), and left lateral (E) views.** Abbreviations: art.clp, articular surface to receive the clinoid process of the basisphenoid; art.ls, articular surface for the laterosphenoid; art.p, articular surface for the parietals; CNV, foramen for cranial nerve V; dp, depression; fac.p, facet for articulation with ventral surface of the base of the posterolateral process of the parietal; fo, fenestra ovalis; gr, groove; hy, groove for hyomandibular branch of the facial nerve; mf, metotic foramen; pop, paraoccipital process; pr.cr, prootic crest; ri1, median ridge on the supraoccipital; ri2, ridge on lateral surface of upper anterior process of the prootic; ri3, ridge on lateral surface of lower anterior process of the prootic; vrop, ventral process of the opisthotic.                     

fading out entirely. A similar ridge is also present in *Shansisuchus shansisuchus* (*Gower & Sennikov, 1996*), but is absent in *Proterosuchus fergusi* (SAM-PK-K10603), *Fugusuchus hejiapanensis* (*Gower & Sennikov, 1996*), *Erythrosuchus africanus* (BP/1/4645), and referred specimens of *Garjainia prima* (e.g., PIN 951/60). The occipital surface of the supraoccipital is concave lateral to the median ridge and becomes flat to gently convex toward the lateral margin of the bone, which is situated at the base of the paroccipital process. The suture between the supraoccipital and the otoccipital is discernible as a low shelf with a dorsolateral–ventromedial orientation. It is not possible to determine whether the supraoccipital contributes to the dorsal border of the foramen magnum or is excluded by a median contact between the otoccipitals. The midline portion of the dorsal surface of the supraoccipital forms a convex articular surface for contact with the ventral surface of the parietals. The lateral half of the dorsal surface of the supraoccipital has a rugose articular facet for the ventral surface of the base of the posterolateral process of the parietal. The anterior surface of the supraoccipital is strongly transversely concave, and articulates dorsolaterally with the prootic and ventrolaterally with the otoccipital.

**Otoccipital (opisthotic + exoccipital).** The left otoccipital is largely intact (Figs. 8 and 9; see also Supplementary Material), but is missing the ventral ramus of the opisthotic and most of the peduncle that forms the lateral wall of the foramen magnum. Only the base of the right otoccipital is preserved, most of the paroccipital process having broken away. The otoccipital forms at least the dorsolateral border of the foramen magnum. The posterior surface of the base of the paroccipital process bears a semi-circular, moderately deep depression on the posterior surface, as in *Garjainia prima* (PIN 2394/5, 951/60), *Garjainia madiba* (*Gower et al., 2014*; BP/1/5760), *Fugusuchus hejiapanensis* (*Gower & Sennikov, 1996*: fig. 4b), *Erythrosuchus africanus* (*Gower, 1997*; UMZC T700), and *Sarmatosuchus otschevi* (*Gower & Sennikov, 1997*). The paroccipital process is mainly posterolaterally directed, but also trends slightly ventrally. The ventral margin of the base of the paroccipital process would originally have been situated above the level of the dorsal margin of the occipital condyle. The occipital surface of the paroccipital process is dorsoventrally convex. The distal end of the process is not dorsoventrally expanded and bears a rounded, asymmetric lateral margin whose lateralmost point lies ventral to the mid-height of the process, resembling the condition in other early archosauriforms (e.g., *Fugusuchus hejiapanensis*: *Gower & Sennikov, 1996*; *Sarmatosuchus otschevi*: *Gower & Sennikov, 1997*; *Garjainia prima*: PIN 2394/5). The dorsal surface of the paroccipital process is rugose. The anterior surface of the base of the paroccipital process articulates with the posterior process of the prootic. The lateral half of the anterior surface of the paroccipital process is weakly dorsoventrally convex, but the distal end of the anterior surface bears a shallow concavity for a loose contact with the posterior surface of the squamosal. The posteroventral margin of the paroccipital process is thin and sharp along its entire length.

The proximal half of the paroccipital process has a thin but well-developed ridge that extends distally from the base of the ventral ramus of the opisthotic and delimits the

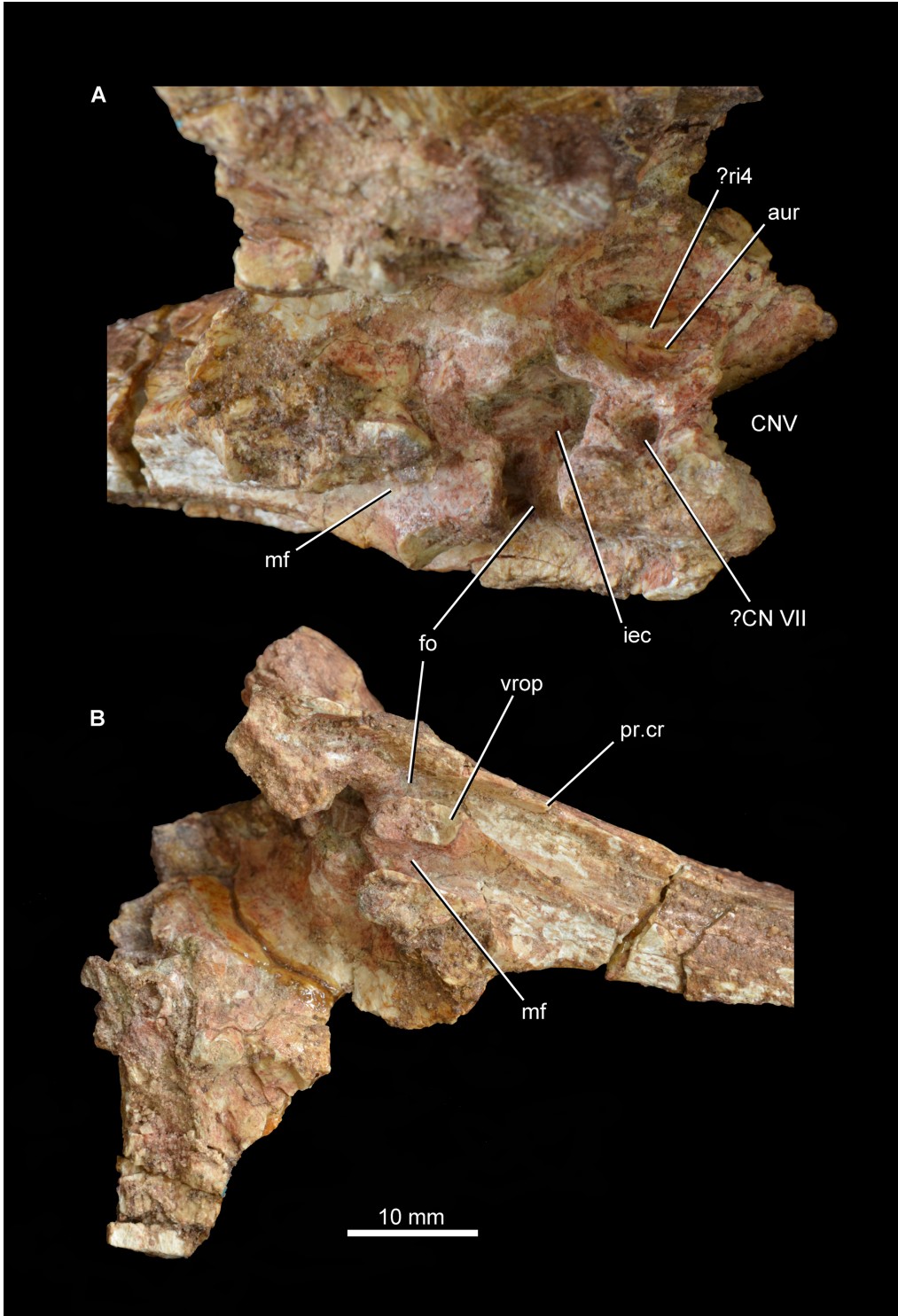

**Figure 9 Partial braincase of *Guchengosuchus shiguaiensis*, IVPP V8808. Medial surface of the prootic (A) and close-up of the braincase in ventral (B) view.** Abbreviations: aur, auricular recess; CNV, foramen for cranial nerve V; ?CN VII, foramen, possibly for passage of cranial nerve VII; fo, fenestra ovalis; iec, inner ear chamber; mf, metotic foramen; pr.cr, prootic crest; ?ri4, possible ridge dividing the auricular recess; vrop, ventral process of the opisthotic.

stapedial groove posteromedially. Only the base of the ventral ramus of the opisthotic, which divides the fenestra ovalis from the anteroposteriorly narrower metotic foramen, is preserved. The ventral ramus of the opisthotic expands slightly anteroposteriorly toward its ventral extremity. The facets for reception of the basioccipital are not preserved.

**Prootic.** The left prootic is almost completely preserved (Figs. 8 and 9; see also Supplementary Material), but lacks the distal tip of the posterior process and most of the lower anterior process that would border the opening for the passage of the trigeminal nerve (cranial nerve V). Only a severely damaged portion of the right prootic is preserved. The posterior process of the prootic tapers posteriorly and participates in forming the base of the paroccipital process, articulating with the anterior surface of the otoccipital. The lateral surface of the posterior process is dorsoventrally convex. A well-developed prootic crest extends posteriorly from the anteroventral corner of the prootic and forms the anterolateral wall of the stapedial groove. The prootic crest decreases in height posteriorly, and eventually merges into the anterior surface of the paroccipital process. The lateral surface of the prootic crest has a ventrally curved groove that is filled with matrix and probably accommodated the hyomandibular branch of the facial nerve (cranial nerve VII), as in other basal archosauriforms (e.g., *Sarmatosuchus otschevi*: *Gower & Sennikov, 1997*; *Fugusuchus hejiapanensis*: *Gower & Sennikov, 1996*). Immediately dorsal to this groove is a second groove that curves gently dorsally and is anteriorly directed toward the border of the foramen for the passage of the trigeminal nerve. A corresponding groove is present in *Sarmatosuchus otschevi* (*Gower & Sennikov, 1997*).

The prootic forms the posterodorsal, posterior, and posteroventral borders of the trigeminal foramen, and the upper anterior process of the prootic possesses on its dorsal surface a deeply recessed articular surface to receive the laterosphenoid. The lateral surface of the upper anterior process of the prootic has a thin and very low posteroventrally directed ridge, as in *Shansisuchus shansisuchus* (*Gower & Sennikov, 1996*). A thin, sharp, posteroventrally aligned ridge is also present on the lateral surface of the lower anterior process of the prootic. This ridge curves slightly ventrally and may represent the upper limit of the area of origination of the protractor pterygoideus muscle (*Gower & Sennikov, 1996*), showing a very similar morphology and position to an equivalent feature in *Sarmatosuchus otschevi* (*Gower & Sennikov, 1997*), *Fugusuchus hejiapanensis* (*Gower & Sennikov, 1996*), and *Garjainia prima* (*Gower & Sennikov, 1996*; PIN 2394/5). The ventral surface of the lower anterior process of the prootic bears a rugose, subrectangular articular surface to receive the clinoid process of the parabasisphenoid. The broad separation between the upper and lower anterior processes of the prootic indicates that the laterosphenoid participated extensively in the border of the trigeminal foramen, as occurs in several other early archosauriforms (e.g., *Proterosuchus fergusi*: *Cruickshank, 1972*, BP/1/3993; *Proterosuchus goweri*: NM QR 880; *Proterosuchus alexanderi*: NM QR 1484; *Sarmatosuchus otschevi*: *Gower & Sennikov, 1997*, PIN 2865/68; *Erythrosuchus africanus*: *Gower, 1997*, UMZC T700; *Shansisuchus shansisuchus*: *Gower & Sennikov, 1996*, fig. 6b). By contrast, the lower and upper anterior processes of the prootic closely approach each other anterior to the trigeminal foramen in

 

*Fugusuchus hejiapanensis* (*Gower & Sennikov, 1996*) and contact each other in *Garjainia prima* and probably a referred specimen of *Erythrosuchus africanus* (*Gower & Sennikov, 1996*; *Gower, 1997*). This condition results in the laterosphenoid being nearly or completely excluded from the border of the trigeminal foramen.

The medial surface of the prootic possesses a complex topography (Fig. 9A; see also Supplementary Material). The fenestra ovalis opens medially into the deep inner ear chamber. The medial surface of the lower anterior process has a large circular foramen that opens ventromedially and probably represents the passage of the cranial facial nerve. On the medial surface of the base of the upper anterior process is a large and moderately deep subcircular fossa that represents the auricular recess. The recess seems to be subdivided by an anterodorsally oriented ridge, but this might be an artefact of damage to the bone surface in the dorsal half of the recess. The morphology of the area that bears the auricular recess and internal ear chamber is consistent with that observed in *Erythrosuchus africanus* (*Gower, 1997*).

**Pterygoid.** The left pterygoid was originally relatively complete (Fig. 4B; *Peng, 1991*: pl. 1.8; see also Supplementary Material), but now only a fragment of the anterior process is preserved (Fig. 10). This includes a medially placed, transversely compressed, dorsally-extending sheet, which expands in dorsoventral height anteriorly; a near-horizontal, dorsoventrally compressed sheet that expands transversely toward its anterior end; and a small part of the sheet of bone that would have connected the horizontal sheet to the ventrolateral process. The ventral surface of the anterior process of the pterygoid is transversely convex and the dorsal surface transversely concave. No teeth are visible on the preserved fragment, consistent with the statement of *Peng (1991)* that palatal teeth were absent. The pterygoids of *Erythrosuchus africanus* (*Gower, 2003*), *Uralosaurus magnus* (PIN 2973/70), *Shansisuchus shansisuchus* (*Young, 1964*, fig. 15a) and the holotype of *Garjainia prima* (PIN 2394/5) also lack palatal teeth, but a referred specimen of *Garjainia prima* possesses two anteroposteriorly short rows of palatal teeth that extend onto the palatine (PIN 951/18).

The portions of the pterygoid of *Guchengosuchus shiguaiensis* that are no longer preserved (*Peng, 1991*, pl. 1.8) resembled the corresponding parts of the bone in several other early archosauriforms. The anterior process of the pterygoid was transversely broad along its entire length, as occurs in the vast majority of non-pseudosuchian archosauromorphs (*Ezcurra, 2016*). The ventrolateral process was distinctly posteriorly oriented, as in *Proterosuchus fergusi* (RC 59, SAM-PK-11208), *Proterosuchus alexanderi* (NM QR 1484), "*Chasmatosaurus*" *yuani* (IVPP V4067), *Sarmatosuchus otschevi* (PIN 2865/68), *Garjainia prima* (PIN 2394/5), and *Erythrosuchus africanus* (NHMUK PV R3592).

**Mandible.** The right articular and the posterior part of the right surangular are present and preserved in articulation (Fig. 11; see also Supplementary Material). No fragments of the angular appear to be present. The anterior end and posterior tip of the surangular are missing and the ventral margin is damaged over most of its length. The medial surface

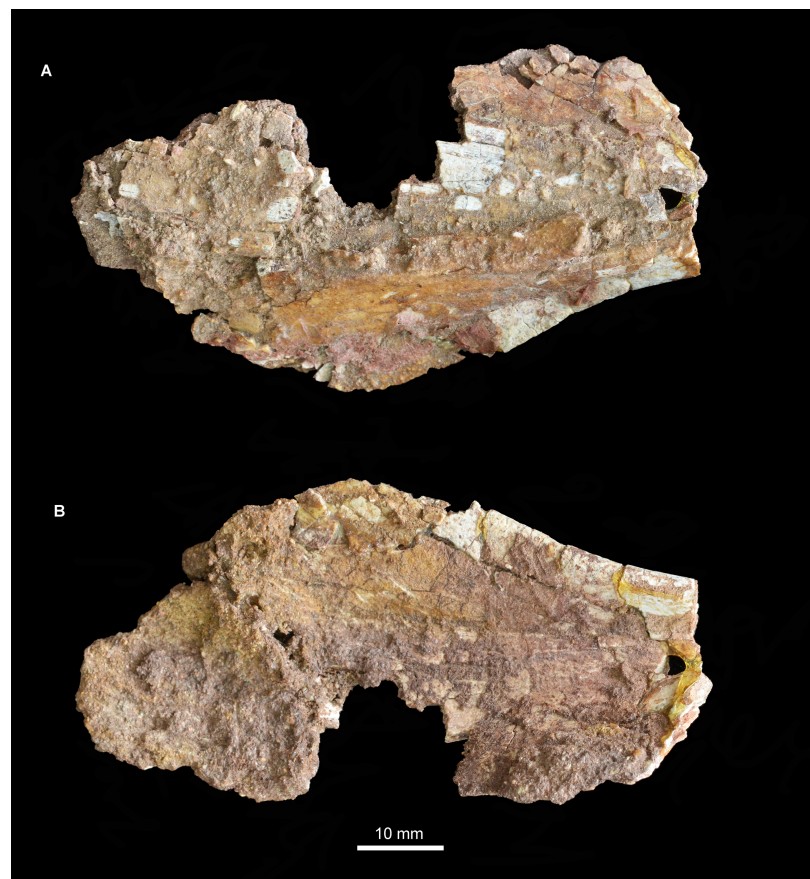

**Figure 10 Fragment of left pterygoid of *Guchengosuchus shiguaiensis*, IVPP V8808, in ventral (A), and dorsal (B) views.**

of the surangular, which forms the lateral wall of the adductor fossa, is mostly obscured by sediment. The lateral surface of the surangular is relatively flat but is badly cracked, with no clear foramina visible. The dorsal margin of the surangular anterior to the glenoid is gently convex in lateral view and laterally expanded to form a shelf that overhangs the rest of the lateral surface of the surangular, resembling the condition in some other archosauriforms (e.g., "*Chasmatosaurus*" *yuani*: IVPP V4067; *Euparkeria capensis*: SAM-PK-5867; *Youngosuchus sinensis*: IVPP V3239). This shelf is better developed and has a more convex lateral edge in *Garjainia prima* (PIN 2394/5, 951/46), *Erythrosuchus africanus* (NHMUK PV R3592), *Shansisuchus shansisuchus* (*Young, 1964*), proterochampsids (e.g., *Chanaresuchus bonapartei*: PULR 07), and some pseudosuchians (e.g., *Tarjadia ruthae*: *Ezcurra et al., 2017*; *Riojasuchus tenuisceps*: *Baczko & Desojo, 2016*). The dorsal surface of this shelf is very gently concave transversely, being dorsoventrally thickened at its lateral and medial margins. The soft tissue that covered this concavity received the ventral surface of the lower temporal bar and its associated soft tissue during full occlusion of the lower jaw. More posteriorly the surangular forms the lateral margin of the glenoid and laterally overlaps the articular. The anterior border of the glenoid is not very well preserved, but seems to be low with a rugose dorsal surface.

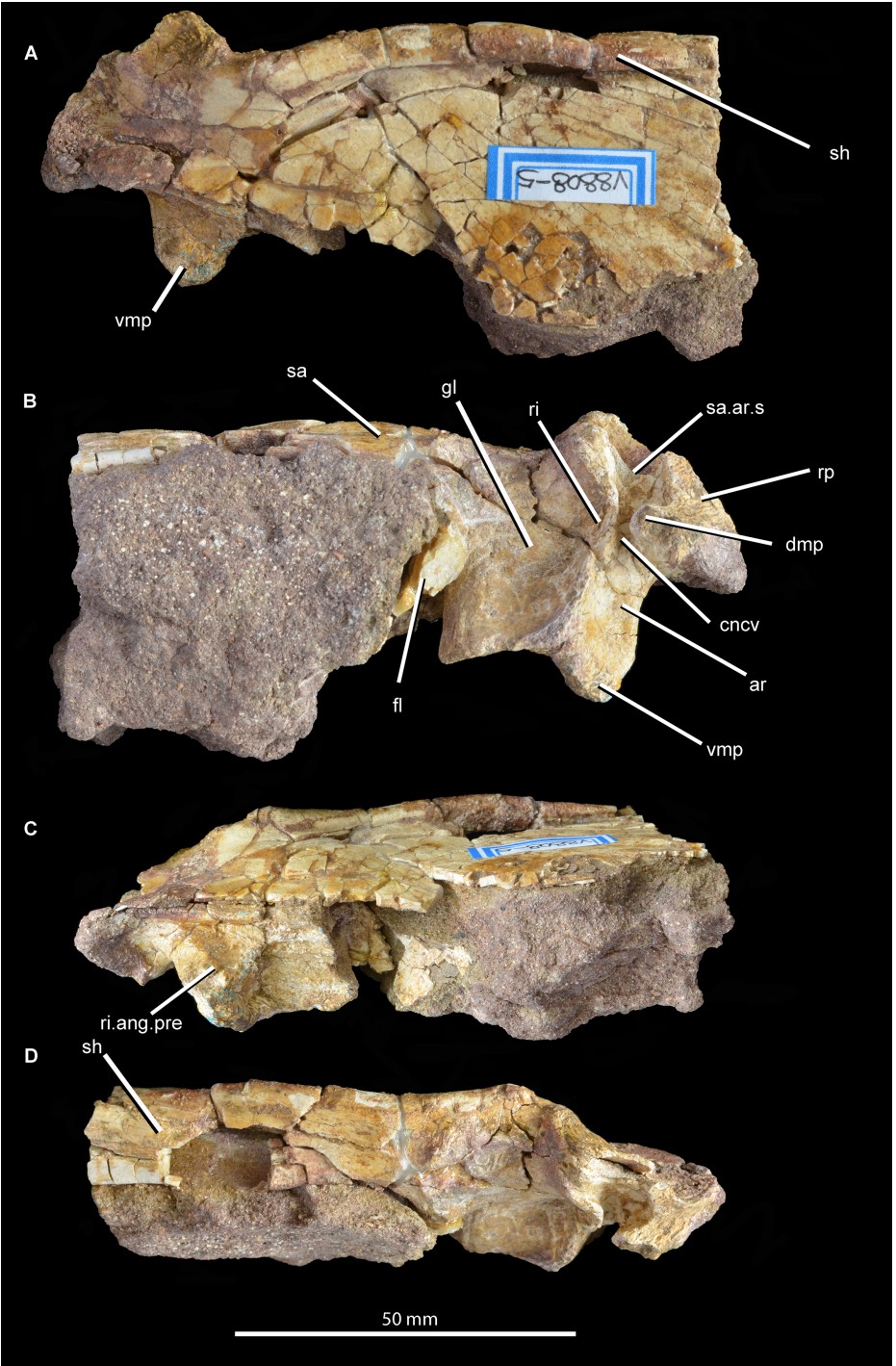

**Figure 11 Posterior right mandible of *Guchengosuchus shiguaiensis*, IVPP V8808, in lateral (A), medial (B), ventral (C), and dorsal (D) views.** Abbreviations: ar, articular; cncv, deep concavity separating the glenoid fossa from the retroarticular process; dmp, dorsomedial projection of the articular; fl, ventromedially directed flange of the surangular, anteriorly bordering the articular; gl, glenoid fossa; ri, ridge forming the posterior border of the glenoid; ri.ang.pre, ridge on ventral surface of articular separating the articular facets for the angular and the prearticular; rp, retroarticular process; sa, surangular; sa.ar.s, suture between the surangular and the articular; sh, shelf on dorsal margin of surangular; vmp, ventromedial process of the articular.

The posterior border of the glenoid is considerably higher than the anterior border, and is formed primarily by the articular but incorporates a small contribution from the surangular. The suture between the surangular and the articular is very clear posterior to the glenoid fossa, but is not discernible on the glenoid articular surface.

The posteriormost part of the surangular is broken away, but would have formed at least part of the lateral surface of the retroarticular process. Most of the medial side of the surangular is obscured by sediment, but a ventromedially directed flange is partially visible. The posterior margin of this flange would have articulated with the articular and formed the anterior margin of the glenoid.

The articular forms most of the glenoid fossa and the retroarticular process. Almost all of the articular is artificially compressed and displaced, such that the glenoid fossa faces mainly medially. As a result of this distortion and the loss of the angular, the medial projection of the articular is artificially well exposed in lateral view. The glenoid fossa is transversely expanded and saddle-shaped, with a smaller lateral concavity separated by a low convexity from a larger and deeper medial concavity. The size discrepancy between the concavities suggests that the medial ventral condyle of the quadrate was considerably transversely wider than the lateral ventral condyle. By contrast, the ventral condyles of the quadrate are subequal in width in *Garjainia prima* (PIN 2394/5). A raised, posteriorly convex ridge of bone forms the posterior border of the glenoid. This ridge is separated by a deep, smooth concavity from the retroarticular process, as occurs in *Proterosuchus fergusi* (BSPG 1934 VIII 514), *Garjainia prima* (PIN 2394/5), referred specimens of *Garjainia prima* (PIN 951/33), and *Garjainia madiba* (NM QR 3051). The retroarticular process is relatively short and not upturned at its distal end, resembling the condition in *Erythrosuchus africanus* (*Gower, 2003*) and *Garjainia prima* (PIN 2394/5). The medial surface of the retroarticular process displays the broken base of a dorsomedial projection. Behind the medial part of the posterior margin of the glenoid fossa is a ventromedial process that extends a very short distance medial to the medial edge of the fossa in dorsal view. An anteroposteriorly extending ridge on the ventral surface of the articular separates the facet for the angular on the ventrolateral surface from the facet for the prearticular on the ventromedial surface. The foramen for the passage of the *chorda tympani* is not preserved. The lateral surface of the retroarticular process is dorsoventrally concave and would have been covered by the posterior tip of the surangular.

**Vertebrae.** *Peng (1991)* originally listed three cervical, six dorsal and three caudal vertebrae as present in the type specimen. Only two cervical vertebrae (the only vertebrae that were figured by *Peng, 1991*, fig. 5, pl. 2.1, 2.2) are now present (Figs. 12–15; see also Supplementary Material), along with a poorly preserved dorsal vertebra and one other vertebral fragment. The two cervical vertebrae are postaxial but probably from a relatively anterior position within the neck. They were figured in articulation by *Peng (1991)* and fit together well. In both vertebrae (as well as in the dorsal vertebra) the neurocentral suture appears fused, suggesting that the holotype of *Guchengosuchus shiguaiensis* does not represent a juvenile individual. Preservation is relatively poor.

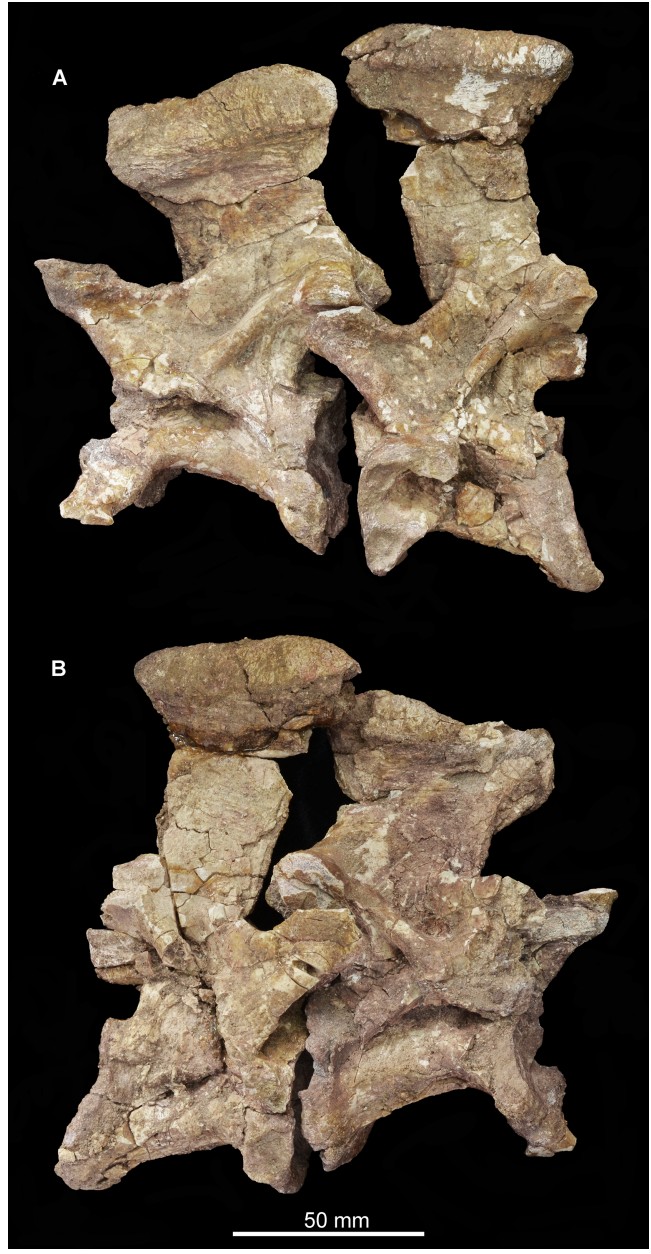

50 mm

**Figure 12** Articulated cervical vertebrae of *Guchengosuchus shiguaiensis*, IVPP V8808, in left lateral (A) and right lateral (B) views.

The first cervical ("cervical a"; Figs. 12–14) has a centrum that is anteroposteriorly longer than dorsoventrally high, its length being approximately 150% of its height, resembling the condition in *Teyujagua paradoxa* (UNIPAMPA 653 cast), *Proterosuchus fergusi* (BSPG 1934 VIII 514, SAM-PK-11208), and *Euparkeria capensis* (SAM-PK-5867). By contrast, the cervical centra are only slightly longer than tall in *Sarmatosuchus otschevi* (PIN 2865/68) and *Garjainia prima* (PIN 2394/5), and considerably anteroposteriorly shorter than tall in *Erythrosuchus africanus* (*Gower, 2003*; SAM-PK-3028) and *Shansisuchus shansisuchus* (*Young, 1964*, fig. 20e). The ventral

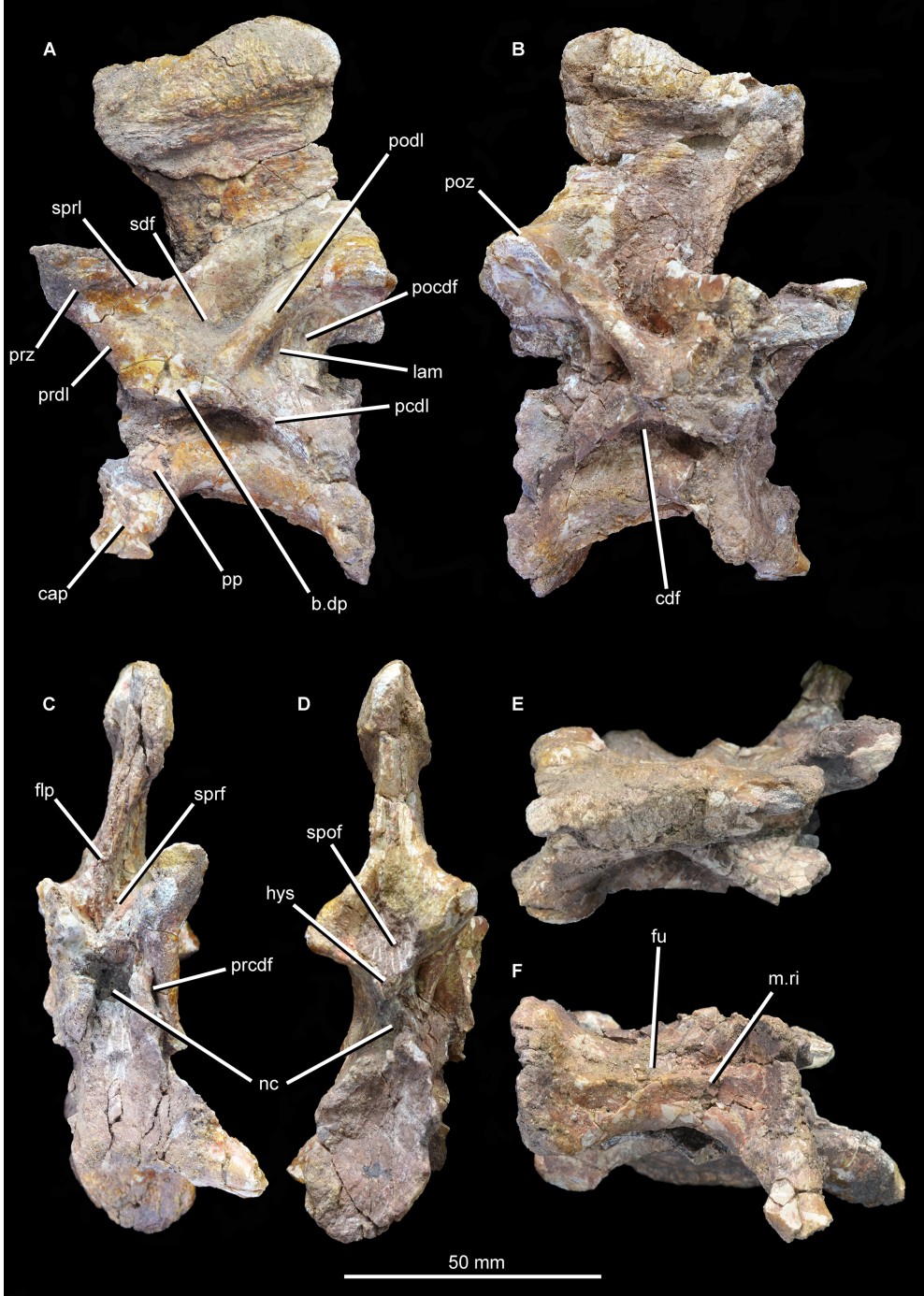

**Figure 13** "Cervical a" of *Guchengosuchus shiguaiensis*, IVPP V8808, in left lateral (A), right lateral (B), anterior (C), posterior (D), dorsal (E), and ventral (F) views. Abbreviations: b.dp, base of the diapophysis; cap, capitulum; cdf, centrodiapophyseal fossa; flp, flange-like projection at the base of the neural spine; fu, furrow; hys, hyposphene; lam, lamina; m.ri, midline ridge; nc, neural canal; pp, para-pophysis; pcdl, posterior centrodiapophyseal lamina; pocdf, postzygapophyseal centrodiapophyseal fossa; poz, postzygopophysis; pp.af, parapophyseal articular facet; podl, postzygodiapopohyseal lamina; prcdf, prezygapophyseal centrodiapophyseal fossa; prdl, prezygodiapopyseal lamina; prz, prezygapophysis; sdf, spinodiapophyseal fossa; spof, spinopostzygapophyseal fossa; sprf, spinoprezygapophyseal fossa; sprl, spinoprezygopophyseal lamina; 3af, third articular facet.

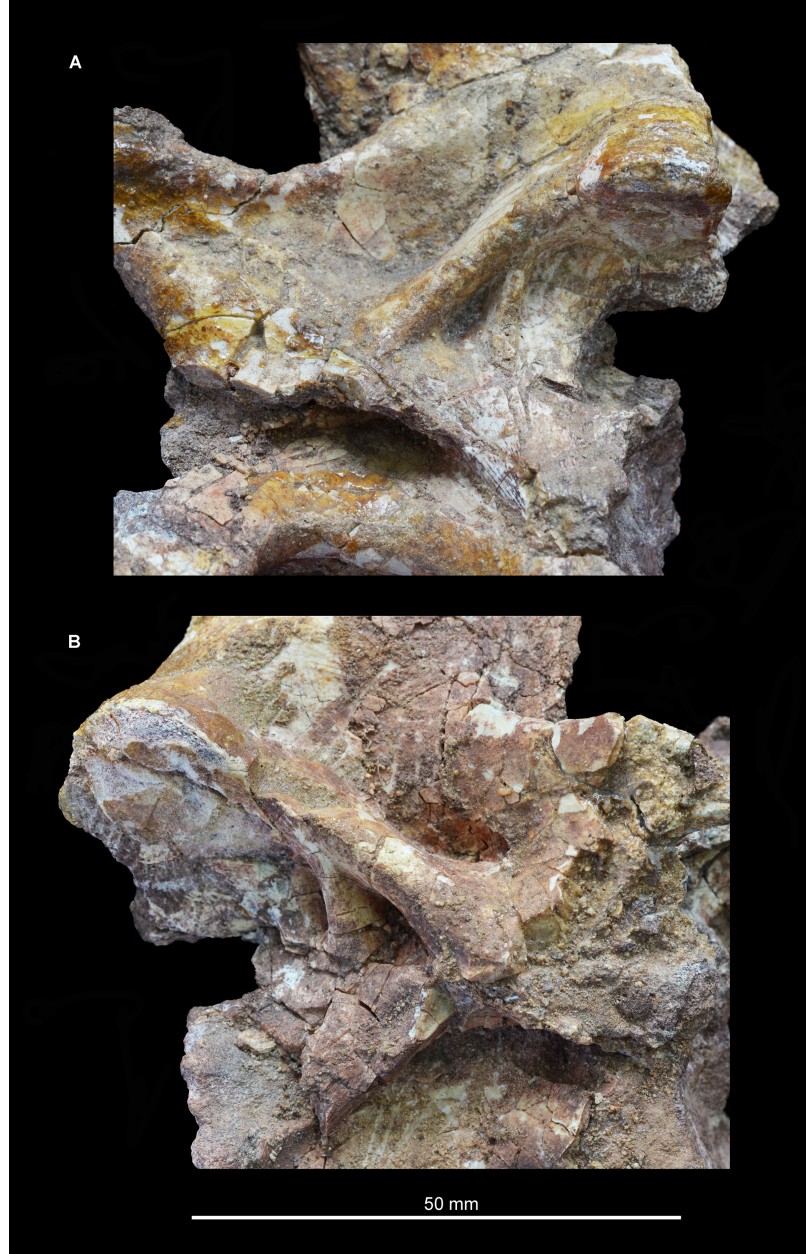

50 mm

**Figure 14** "Cervical a" of *Guchengosuchus shiguaiensis*, IVPP V8808, in close-up left lateral (A) and right lateral (B) views.

margin of the centrum is strongly concave in lateral view. The centrum is parallelogram-shaped in lateral view, with the anterior articular surface being more dorsally positioned than the posterior one, resembling the condition in several other early archosauromorphs (*Ezcurra, 2016*). The anterior and posterior articular surfaces of the centrum have oval outlines (although the anterior articular face is broken along its right lateral margin and damaged dorsally and ventrally), being dorsoventrally deeper than transversely wide, and are deeply concave. The oval outlines of the centra may however be exaggerated by post-mortem transverse compression of the elements. The lateral

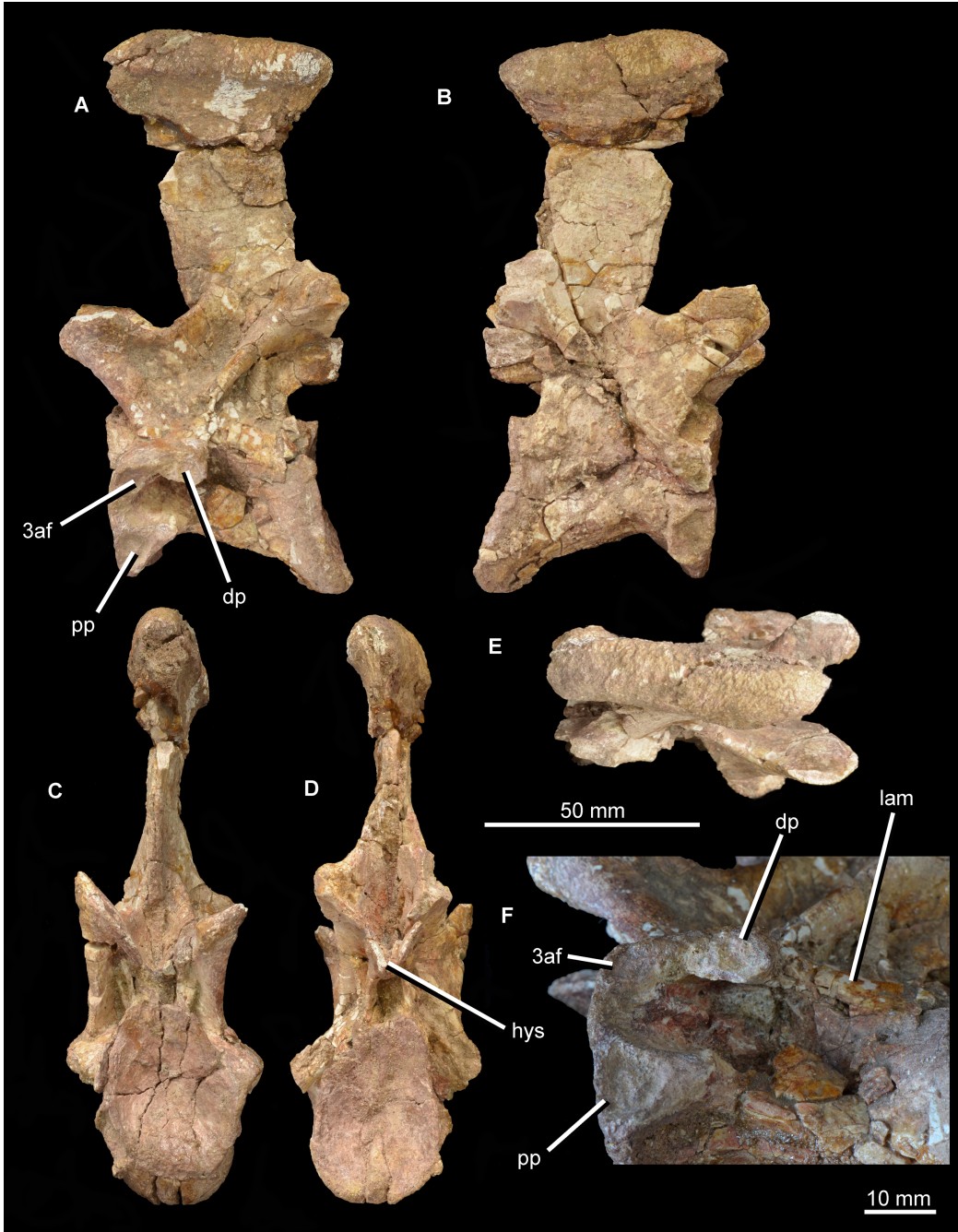

**Figure 15** "Cervical b" of *Guchengosuchus shiguaiensis*, IVPP V8808, in left lateral (A), right lateral (B), anterior (C), posterior (D), dorsal (E), and close-up left lateral (F) views. Abbreviations: dp, diapophysis; hys, hyposphene; lam, lamina; pp, parapophysis; 3af, third articular facet.

surfaces of the centrum are strongly pinched inward, giving the centrum an hourglass-like outline in ventral view. The ventral surface of the centrum is flattened but bears a low midline ridge, lateral to which is a shallow, anteroposteriorly-extending furrow that deepens slightly anteriorly below the parapophysis. A median ventral ridge is also present

in the anterior postaxial centra of *Sarmatosuchus otschevi* (PIN 2865/68), *Erythrosuchus africanus* (BP/1/4680), *Garjainia prima* (PIN 2394/5), *Prolacerta broomi* (BP/1/2675), *Proterosuchus fergusi* (BSPG 1934 VIII 514), and *Euparkeria capensis* (SAM-PK-5867). The lateral surface of the centrum is deeply excavated, forming a deep centrodiapophyseal fossa ventral to the inferred position of the neurocentral suture and posteroventral to the diapophysis. This fossa undercuts the ventral surface of the diapophysis. The parapophysis is broken away on the right side, but on the left side is positioned on the anteroventral corner of the centrum and has fused to the capitulum of the cervical rib. There is no indication on either side of a third articular surface for the rib, although this area of the vertebra is not well preserved.

The diapophysis is broken away on both sides, but its articular facet would have been set at the end of a ventrolaterally directed process, the base of which is placed approximately on the neurocentral suture, just anterior to the midlength of the centrum. The ventral orientation of the diapophyses is probably exaggerated by post-mortem transverse compression. A well-defined and thick postzygodiapopohyseal lamina arches posterodorsally from the base of the diapophysis to connect to the anterior margin of the postzygapophysis. A low posterior centrodiapophyseal lamina extends posteroventrally from the base of the diapophysis onto the posterodorsal portion of the centrum, forming the posterodorsal wall of the centrodiapophyseal fossa described above. The postzygodiapopohyseal lamina and the weakly developed posterior centrodiapophyseal lamina together frame a postzygapophyseal centrodiapophyseal fossa positioned posterior to the diapophysis and anteroventral to the postzygapophysis, as occurs in the cervical and dorsal vertebrae of several other archosauriforms (e.g., *Erythrosuchus africanus*: NHMUK PV R3592; *Gower, 2001*, *2003*). This fossa is divided into anterior and posterior parts by an unusual and likely autapomorphic lamina that extends ventrally from the postzygodiapopohyseal lamina. A low thickening extending from the base of the diapophysis to the underside of the prezygapophysis is in an equivalent position to a prezygodiapophyseal lamina, as in *Tanystropheus longobardicus* (*Wild, 1973*, figs. 52–54), *Protorosaurus speneri* (BSPG 1995 I 5), *Cuyosuchus huenei* (MCNAM 2669), *Erythrosuchus africanus* (NHMUK PV R3592, *Gower, 2003*), *Shansisuchus shansisuchus* (*Young, 1964*, fig. 21), and *Euparkeria capensis* (UMZC T921). The prezygodiapophyseal lamina of *Guchengosuchus shiguaiensis* was probably originally placed further laterally to the base of the prezygapophysis before post-mortem compression. Although poorly preserved in this vertebra, this low thickening forms the anteroventral border of a shallow fossa on the lateral wall of the neural canal. This fossa represents a prezygapophyseal centrodiapophyseal fossa. Just ventral to the base of the neural spine, the lateral surface of the neural arch bears another deep fossa, which is bounded by the postzygodiapophyseal lamina, the weakly developed prezygodiapophyseal lamina, and a spinoprezygapophyseal lamina that extends from the posterior margin of the prezygapophysis onto the lateral surface of the neural spine. This is a spinodiapophyseal fossa and resembles the excavation present immediately lateral to the base of the neural spine in the postaxial cervical vertebrae of several other archosauromorphs, such as *Protorosaurus speneri* (BSPG 1995 I 5, cast of WMsNP47361),

*Eorasaurus olsoni* (PIN 156/108-110), *Proterosuchus alexanderi* (NM QR1484), *Vonhuenia fredericki* (PIN 1025/11), *Garjainia prima* (PIN 2394/5), *Cuyosuchus huenei* (MCNAM PV 2669), and *Euparkeria capensis* (SAM-PK-5867).

The right prezygapophysis is broken away, but the left one is a large triangular process that extends anterodorsally a substantial distance beyond the anterior margin of the centrum. The articular face of the left prezygapophysis is poorly preserved, but faces dorsomedially at a low angle to the horizontal. The articular facet of the prezygapophysis is oval and almost flat. Between and ventral to the prezygapophyses, the anterior opening of the neural canal is poorly preserved because the dorsal margin of the centrum is broken, but appears to have been subcircular. Posteriorly, the postzygapophyses flare mainly laterally, and project a short distance beyond the centrum; their articular surfaces face ventrolaterally and are sub-circular. Below the postzygapophyses the posterior opening of the neural canal is poorly preserved. A well-developed hyposphene is present ventrally between the postzygapophyses, overhanging the neural canal; anteriorly, the presence of a hypantrum cannot be confirmed due to damage. The hyposphene is formed by medioventral projections of the postzygapophyses and forms the floor of the spinopostzygapophyseal fossa.

The neural spine is anteroposteriorly broad and set above the posterior 80% of the centrum, resembling the condition in *Sarmatosuchus otschevi* (PIN 2865/68), *Teyujagua paradoxa* (UNIPAMPA 653 cast), *Garjainia prima* (PIN 2394/5), some specimens of *Proterosuchus fergusi* (BSPG 1934 VIII 514; SAM-PK-11208) and *Euparkeria capensis* (SAM-PK-5867). The spine is inclined slightly anteriorly, an unusual condition among early archosauromorphs that also occurs in the anterior postaxial cervical vertebrae of *Proterosuchus alexanderi* (NM QR 1484) and some specimens of *Proterosuchus fergusi* (GHG 231). The neural spine expands anteroposteriorly toward its apex, the degree of expansion being greater in the anterior direction. The distal third of the spine also forms a strong transverse expansion, which is strongly rugose and dorsally convex. This rugose transverse expansion is well developed ventrally, extending along the distal third of the neural spine. This feature is also present in several isolated neural spines from the Lower Triassic Rewan Formation of Queensland (*Thulborn, 1979*; QM F10125) and the aphanosaurian avemetatarsalians *Teleocrater rhadinus* (*Nesbitt et al., 2018*) and *Yarasuchus deccanensis* (ISIR 334). Anterior to the base of the spine is a well-developed spinoprezygapophyseal fossa bounded laterally by the prezygapophyses and dorsolaterally by the spinoprezygapophyseal laminae. The ventral part of this fossa is divided into left and right halves by an anterior, flange-like projection of the base of the neural spine. Posteriorly, a well-developed spinopostzygapophyseal fossa is placed between the postzygapophyses and extends onto the ventral half of the neural spine.

The second cervical vertebra ("cervical b"; Figs. 12 and 15) is generally very similar to the first. The low ridge on the ventral surface of the centrum is less well developed. There is no clear bevelling for articulation with an intercentrum (a characteristic that cannot be assessed in "cervical a" because of damage). The diapophysis is completely preserved on the left side, and the articular facet for the rib is set at the end of a short ventrolaterally extending process. The parapophyseal articular facet is larger than the

diapophyseal one, and is situated on the anteroventral corner of the centrum. The parapophysis is not situated on a significant prominence, and in this case is not fused to the capitulum of the rib on either side of the vertebra. On the anterodorsal corner of the centrum, at the same horizontal level as the diapophysis, is a third facet for rib articulation. This condition differs from that observed in other species with three-headed ribs (e.g., *Prolacerta broomi*: BP/1/2675; *Proterosuchus alexanderi*: NM QR 1484; *Chasmatosuchus rossicus*: PIN 2252/381; *Erythrosuchus africanus*: *Gower, 2003*; *Cuyosuchus huenei*: MCNAM PV 2669; *Teleocrater rhadinus*: NHMUK PV R6795), in which the third facet is situated anteroventral to the diapophysis on a thick paradiapophyseal lamina. The third articular facet is separated from the diapophyseal facet by a small non-articular area, but lies adjacent to the anterior margin of the transverse process. A moderately thin lamina extends posteriorly from the base of the third articular facet and subdivides the centrodiapophyseal fossa of the centrum, extending as far posteriorly as does the base of the diapophysis. To our knowledge, this lamina is not present in other non-archosaurian archosauromorphs.

The same set of neural arch fossae and laminae are present as in cervical a, but the prezygapophyseal centrodiapophyseal fossa is better preserved and more clearly developed. The hyposphene is well developed, and what appears to be a flat, oval hypantral facet is present on the base of the right prezygapophysis. The neural spine of cervical b is similar to that of cervical a, but is taller, more vertically oriented, and both absolutely and proportionately narrower anteroposteriorly.

The partial dorsal vertebra (Fig. 16) includes the dorsal part of the posterior articular face of the centrum, as well as the neural spine, transverse processes, and zygapophyses. This vertebra is probably an anterior dorsal because of the presence of strongly laterally projecting transverse processes, the anterodorsal orientation of the prezygapophyses and the lateroventrally facing facets of the postzygapophyses. The neurocentral suture is fully closed. The dorsal part of the posterior articular face of the centrum is gently concave, and above it there is a subcircular neural canal. The diapophyseal facet is set at the end of an elongate, laterally directed transverse process. A well-developed posterior centrodiapophyseal lamina extends posteroventrally from the diapophysis, as also occurs in the non-crocopod archosauromorphs *Tanystropheus longobardicus* (*Wild, 1973*: figs. 52–54) and *Protorosaurus speneri* (BSPG 1995 I 5), and the early archosauriforms *Cuyosuchus huenei* (MCNAM 2669), *Garjainia prima* (PIN 2394/5-16), *Erythrosuchus africanus* (NHMUK PV R3592; *Gower, 2003*), *Shansisuchus shansisuchus* (*Young, 1964*: fig. 21), and *Euparkeria capensis* (UMZC T921). A paradiapophyseal lamina, if one was originally present, has broken away. The posterior centrodiapophyseal lamina extends longitudinally along the entire ventral surface of the transverse process and gives it a subtriangular cross-section, resembling the condition in *Garjainia prima* (PIN 2394/5-16). The facet of the diapophysis is subtriangular and strongly concave. A deep postzygapophyseal centrodiapophyseal fossa is present posterior to the diapophysis; it seems likely that prezygopophyseal centrodiapophyseal and centrodiapophyseal fossae were also present, although the area they would have occupied is damaged. A deep spinodiapophyseal fossa is present dorsal to the transverse

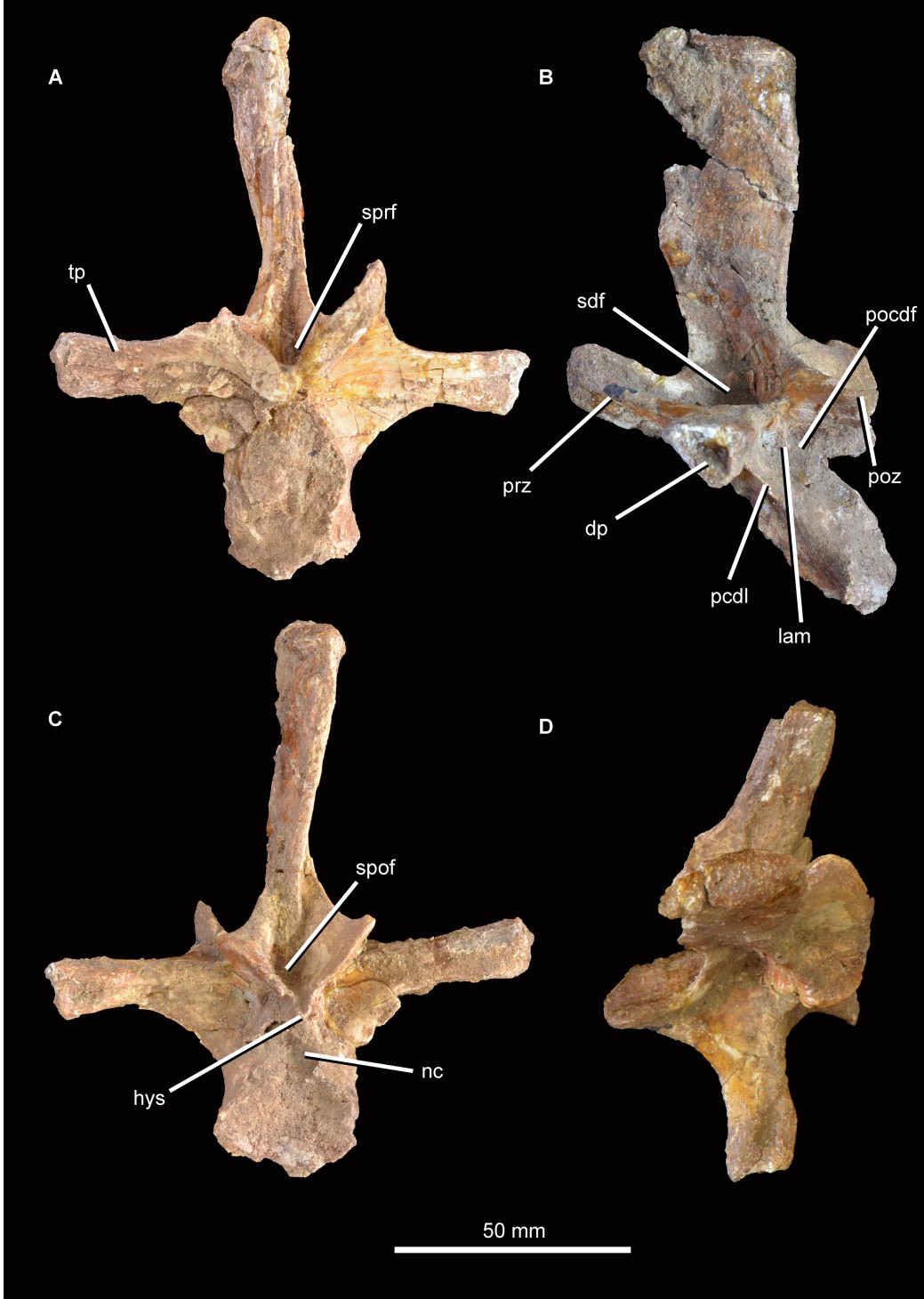

**Figure 16 Partial dorsal vertebra of *Guchengosuchus shiguaiensis*, IVPP V8808, in anterior (A), left lateral (B), posterior (C), and dorsal (D) views.** Abbreviations: dp, diapophysis; hys, hyposphene; lam, accessory lamina; nc, neural canal; pcdl, posterior centrodiapophyseal lamina; pocdf, postzygapophyseal centrodiapophyseal fossa; poz, postzygapophysis; prz, prezygapophysis; sdf, spinodiapophyseal; spof, spinopostzygapophyseal fossa; sprf, spinoprezygapophyseal fossa; tp, transverse process.

process, resembling the condition in *Erythrosuchus africanus* (NHMUK PV R3592), but contrasting with the distinctly shallower fossa present in *Garjainia prima* (PIN 2394/5-16) and *Garjainia madiba* (BP/1/7135). Spinoprezygapophyseal and spinopostzygapophyseal fossae are present anterior and posterior to the base of the neural spine in *Guchengosuchus shiguaiensis*. An accessory, posteroventrally oriented lamina divides the postzygapophyseal centrodiapophyseal fossa, as also occurs in the cervical vertebrae. The prezygapophysis projects anterodorsally and its articular surface faces dorsomedially at around 45° to the horizontal; the postzygapophysis faces ventrolaterally at a similar angle. A well-preserved hyposphene with the same morphology as those of the cervical vertebrae is present posteriorly, resembling the condition in the dorsal vertebrae of *Erythrosuchus africanus* (NHMUK PV R3592) and *Sarmatosuchus otschevi* (PIN 2865/68). The neural spine is elongate and anteroposteriorly narrow; it widens slightly anteroposteriorly toward its apex, but considerably less than occurs in the cervicals. The morphology of this neural spine closely resembles that of the cervico-dorsal vertebrae of *Garjainia prima* (PIN 2394/5-16). The neural spine apex is rugose and slightly expanded transversely.

A fragment of an additional vertebra (Fig. 17) includes the concave articular face of a centrum, and a small part of the neural canal and arch, but provides no useful anatomical data. A ventral keel is present as in the other vertebrae.

**Ribs.** Four partial ribs are preserved (Fig. 18), although in each case the distal portion is lacking. A left cervical rib (Figs. 18A–18C) has a well-separated capitulum and tuberculum. The articular surface of the capitulum has broken away, but the process is flattened from anteroventral to posterodorsal. The tuberculum is an elongate process bearing a triangular articular facet. At the intersection of the capitulum and tuberculum the rib is drawn out into a tapering anterior process, as occurs in other archosauromorphs (*Ezcurra, 2016*). The shaft of the rib is slightly laterally convex along its length and has a T-shaped cross section produced by grooves lying along the ventro- and dorsomedial edges.

Two "pectoral" ribs are preserved. One of these is clearly three-headed (Figs. 18I and 18J), but the probable third head of the second (Figs. 18F–18H) is damaged at its putative articular end. At least one three-headed "pectoral" rib also occurs in *Prolacerta broomi*, proterosuchids, *Vonhuenia fredericki*, *Chasmatosuchus rossicus*, *Sarmatosuchus otschevi*, *Cuyosuchus huenei*, other erythrosuchids, aphanosaurs, and some paracrocodylomorphs (*Nesbitt, 2011*; *Ezcurra, 2016*; *Nesbitt et al., 2018*). The less complete of the two pectoral ribs is from the left side and has a broadly separated capitulum and tuberculum. As in the cervical rib, the capitulum is anteroposteriorly compressed whereas the tuberculum has a nearly circular cross section. A third process extends proximally from the base of the capitulum and is anteroposteriorly compressed, but its articular end is incomplete. The rib shaft is strongly T-shaped at its base, the lateral margin of the shaft being symmetrically expanded anteriorly and posteriorly. In this region the lateral surface of the shaft is gently concave. More distally the rib acquires a nearly oval cross-section with a convex lateral surface, but the posterior margin of the lateral surface is still drawn out slightly posteriorly.

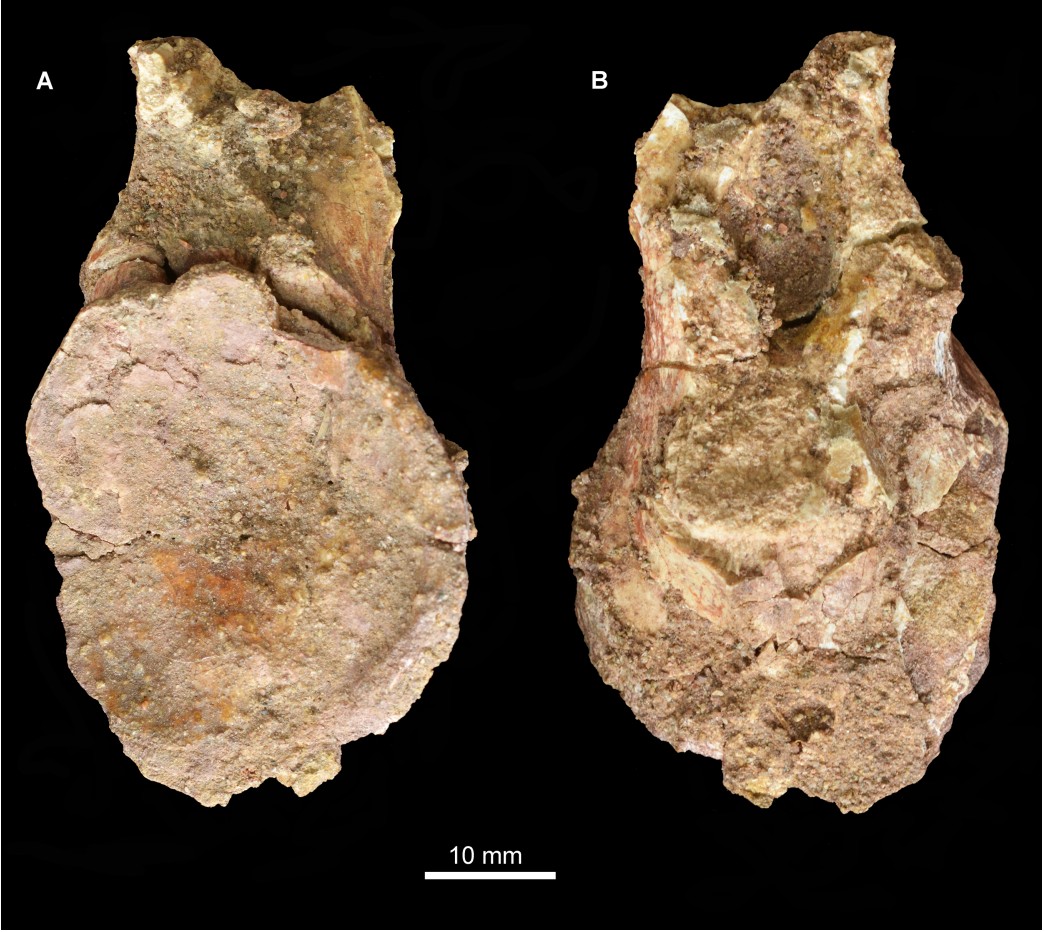

**Figure 17 Vertebral fragment, *Guchengosuchus shiguaiensis*, IVPP V8808.** Because of the incompleteness of the specimen, the orientations of views (A) and (B) are uncertain.

The second pectoral rib is similar to the first (Figs. 18F–18H) but likely belongs to the right side, although the proximal end is not well preserved. The capitulum is again anteroposteriorly compressed, whereas the tuberculum has an oval cross section. A very well-developed third process extends proximally from the base of the capitulum, and is anteroposteriorly compressed with an oval articular surface. More distally, the anterolateral surface of the proximal end of the shaft bears an anteromedially projecting flange, and the posterior surface of the shaft is grooved. Toward its distal end the shaft has an almost oval cross section, the posterior margin of the lateral surface being very slightly drawn out posteriorly.

The fourth rib (Figs. 18D and 18E) may be from the middle or posterior dorsal region and is from the right side. The capitulum is much longer than the tuberculum and is anteroposteriorly compressed, whereas the tuberculum ends in a subcircular articular surface. The capitulum and tuberculum are connected to one another at their bases by a thin web of bone. The shaft is T-shaped, with well-developed grooves on the anterior and posterior surfaces and a flattened lateral surface.

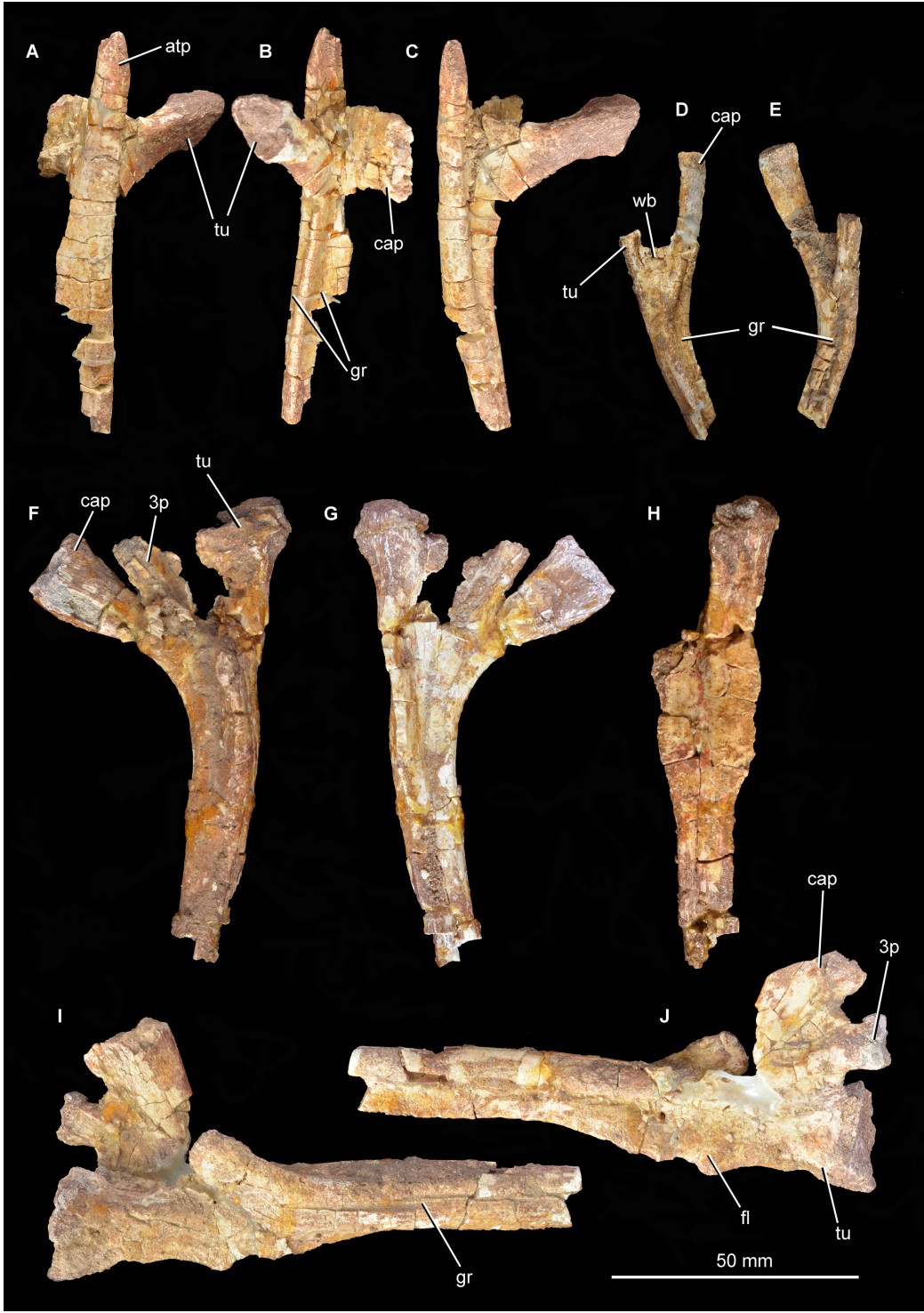

**Figure 18 Partial ribs, *Guchengosuchus shiguaiensis*, IVPP V8808.** Left cervical rib in lateral (A), medial (B), and dorsal (C) views. Right dorsal rib in anterior (D) and posterior (E) views. Left "pectoral" rib in anterior (F), posterior (G), and lateral (H) views. Right "pectoral" rib in anterior (I) and posterior (J) views. Abbreviations: atp, anterior process; cap, capitulum; fl, flange; gr, groove; tu, tuberculum; wb, web of bone between the capitulum and tuberculum; 3p, third process of the rib.

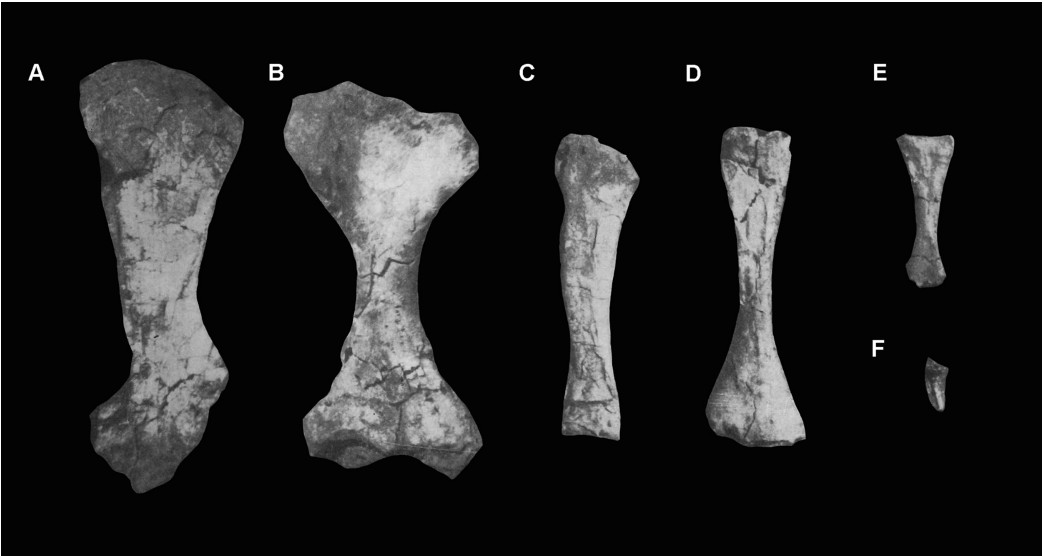

**Figure 19 Non-vertebral postcranial elements of *Guchengosuchus shiguaiensis*, IVPP V8808, as originally preserved and figured by *Peng (1991)*.** All specimens are now missing. Right scapula in lateral view (A), right humerus in anterior view (B), ulna (C), radius (D), metatarsal (E), and ungual phalanx (F). No scale bars were presented in the original figures. *Peng (1991)* did provide reduction factors (e.g., *x*½) for individual bones in his plates; however, the accuracies of these are unclear. As such, the present figure should not be used to estimate relative proportions of individual bones.

**Shoulder girdle (currently lost).** The anterior margin of the scapular blade (*Peng, 1991*, fig. 7, pl. 2.3; Fig. 19A; see also Supplementary Material) is strongly concave in lateral view whereas the posterior margin is almost straight, as also occurs in *Euparkeria capensis* (SAM-PK-5867), *Garjainia prima* (PIN 2394/5; *Huene, 1960*, plate 14, fig. 10), *Erythrosuchus africanus* (*Gower, 2003*; NHMUK PV R3762a), and *Shansisuchus shansisuchus* (*Young, 1964*, fig. 26a). By contrast, in *Prolacerta broomi* (BP/1/2575), *Proterosuchus alexanderi* (NM QR 1484), "*Chasmatosaurus*" *yuani* (IVPP V2719), *Sarmatosuchus otschevi* (PIN 2865/68), and *Cuyosuchus huenei* (MCNAM 2669) the anterior margin of the scapular blade is convex and the posterior margin is posterodistally directed. *Peng (1991)* figured a low, posteriorly oriented tuberosity on the posterior margin of the scapular blade which is visible in his line drawing (Fig. S2), although it is less apparent in his photograph (Fig. 19A). This tuberosity is positioned at the level of the minimum anteroposterior width of the scapular blade and closely resembles in morphology and position the thin, vertical ridge present in *Erythrosuchus africanus* (*Gower, 2003*) and *Shansisuchus shansisuchus* (*Young, 1964*, fig. 26). *Nesbitt (2011)* interpreted this ridge as possibly associated with the origin of the scapular head of the *M. triceps*. This tuberosity is absent in *Proterosuchus alexanderi* (NM QR 1484), *Garjainia prima* (PIN 2394/5), *Garjainia madiba* (BP/1/7152), and *Sarmatosuchus otschevi* (PIN 2865/68-37). A similar tuberosity or ridge is present at or very close to the base of the supraglenoid lip in several archosauriforms (e.g., *Halazhaisuchus qiaoensis*: *Sookias et al., 2014*; *B. kupferzellensis*: *Gower & Schoch, 2009*), being therefore more proximally placed than in *Guchengosuchus shiguaiensis* and *Erythrosuchus africanus*.

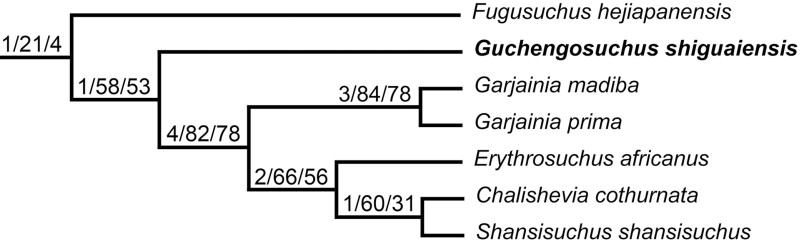

**Figure 20 Phylogenetic relationships within Erythrosuchidae.** Numbers above branches are Bremer support values and absolute and GC bootstrap frequencies.

**Forelimb (currently lost).** The humerus (*Peng, 1991*, fig. 8A, B; pl. 2.4; Fig. 19B; see also Supplementary Material) has strong, symmetrical transverse expansions at the proximal and distal ends, and appears to have a robust, well-developed deltopectoral crest. The ulna (*Peng, 1991*, fig. 8C, pl. 2.5; Fig. 19C; see also Supplementary Material) is very similar to that of *Garjainia madiba* (BP/1/6232r), including in the presence of a squared-off distal end in lateral view. The distal end of the radius (*Peng, 1991*, fig. 8D, pl. 2.6; Fig. 19D; see also Supplementary Material) is strongly expanded anteroposteriorly, as in *Garjainia prima* (*Huene, 1960*), *Erythrosuchus africanus* (*Gower, 2003*; SAM-PK-905), *Shansisuchus shansisuchus* (*Young, 1964*), and the pseudosuchian *Riojasuchus tenuisceps* (PVL 3828).

**Metapodial (currently lost).** A bone interpreted as a non-ungual phalanx by *Peng (1991*, pl. 2.7; Fig. 19E; see also Supplementary Material*)* is relatively large with respect to the other postcranial bones and is elongate relative to its width, indicating that it represents a metapodial. Little information can be derived from the single photograph presented by *Peng (1991)*. An ungual phalanx was also figured by *Peng (1991*, pl. 2.8; Fig. 19F*)*.

## Phylogenetic position

Our phylogenetic analysis recovered 27 MPTs of 3,585 steps, with a consistency index of 0.2519 and a retention index of 0.6477. As in a previous analysis by *Ezcurra (2016)*, *Guchengosuchus shiguaiensis* was recovered within Erythrosuchidae, as the sister-taxon of a clade composed of the genus *Garjainia* and the Middle Triassic species *Erythrosuchus africanus* + (*Shansisuchus shansisuchus* + *Chalishevia cothurnata*) (Fig. 20). The Chinese archosauriform *Fugusuchus hejiapanensis* was found to be the sister taxon of all the other erythrosuchids. None of the character states recovered in the analysis as erythrosuchid synapomorphies could be coded as present or absent in *Guchengosuchus shiguaiensis*, but this species possesses the following synapomorphies of the clade that includes all erythrosuchids to the exclusion of *Fugusuchus hejiapanensis*: maxilla with a low tuberosity delimiting anteriorly the antorbital fenestra and forming a gradual transition with the external surface of the anterior process (character 46: 0 → 1); maxilla with lateroventrally facing neurovascular foramina on the lateral surface of the anterior and horizontal processes and extending ventrally as deep, well-defined grooves (character 53: 0 → 1); maxillary alveolar margin distinctly upturned on the anterior third of

the bone (anterior to the level of the anterior border of the antorbital fenestra if present) (character 70: 0→1); parietal with posterolateral process with a strongly transversely convex dorsal margin elevated from the median line of the posterior margin of the skull roof (character 169: 0→1); and radius with a strongly anteroposteriorly expanded distal end (character 438: 0→1). However, Bremer support for this branch is minimal, and the absolute and GC bootstrap frequencies are 58% and 53%, respectively.

The branch that includes both species of *Garjainia* and Middle Triassic erythrosuchids possesses the following synapomorphies that are absent in *Guchengosuchus shiguaiensis*: maxilla and nasal with a high maxillo-nasal tuberosity, delimiting anteriorly the antorbital fenestra or fossa, forming a distinct change of slope with the external surface of the anterior process (character 46: 1→2); maxilla with an antorbital fossa exposed in lateral view (character 54: 0→1/2); maxilla with a sub-vertical anterior margin of the base of the ascending process (character 58: 1→2); frontal with an only slightly constricted longitudinal canal for the passage of the olfactory tract and no olfactory bulb moulds and distinct semilunate posteromedially-to-anterolaterally oriented ridge on the orbital roof (character 120: 0→1); parietals with a pineal fossa on the median line of their dorsal surface (character 162: 0→1); surangular with a strongly laterally projecting shelf with a strongly convex lateral edge on the dorsolateral surface of the bone (character 286: 2→3); anterior cervical centra with a median, ventral longitudinal keel that extends ventral to the centrum rim in at least one vertebra (character 327: 1→2); and postaxial cervical vertebrae with a shallow, posterolaterally facing fossa ventral to the postzygapophysis (character 335: 0→1). This branch is very well supported, with a Bremer support of 4 and absolute and GC bootstrap frequencies of 82% and 78%, respectively.

Under suboptimal searches, four additional steps are needed to place *Guchengosuchus shiguaiensis* as the sister taxon of all other erythrosuchids (including *Fugusuchus hejiapanensis*), six steps to force its recovery as the sister taxon of *Erythrosuchus africanus* + (*Chalishevia cothurnata* + *Shansisuchus shansisuchus*) and eight steps to make it the sister taxon of *Garjainia*. Finally, six additional steps are needed to position *Guchengosuchus shiguaiensis* outside Erythrosuchidae, either as one of the closest sister taxa of Erythrosuchidae + Eucrocopoda or as the earliest branching member of Eucrocopoda.

## DISCUSSION AND CONCLUSION

The phylogenetic relationships of Erythrosuchidae found in our analysis are congruent with those previously recovered by *Ezcurra (2016)* and *Ezcurra et al. (2018)*. The interrelationships within Erythrosuchidae are relatively robust and do not generate substantial ghost lineages. By contrast, ghost lineages longer than 3 million years are common among other Permo-Triassic archosauromorph clades, such as rhynchosaurs and allokotosaurs (*Nesbitt et al., 2015*; *Ezcurra, 2016*; *Sengupta, Ezcurra & Bandyopadhyay, 2017*). The current potential for direct morphological comparisons between the oldest and earliest branching erythrosuchids is highly limited, mainly because of the fragmentary condition of the holotype and only known specimen of *Guchengosuchus*

*shiguaiensis*. By contrast, the hypodigms of the Early Triassic erythrosuchids *Fugusuchus hejiapanensis* and *Garjainia prima* include fairly complete skulls (*Ochev, 1958*; *Cheng, 1980*), although the whereabouts of the type specimen of the former are currently unknown (*Ezcurra, 2016*). The most striking putative difference between *Guchengosuchus shiguaiensis* on the one hand, and *Fugusuchus* and *Garjainia* on the other, lies in *Peng (1991)* interpretation of *Guchengosuchus shiguaiensis* as possessing a secondary antorbital fenestra, like the more derived Middle Triassic erythrosuchids *Shansisuchus shansisuchus* and *Chalishevia cothurnata*. Phylogenetic relationships within Erythrosuchidae indicate that if a secondary antorbital fenestra was actually present in *Guchengosuchus shiguaiensis*, as proposed by *Peng (1991)*, this feature must have evolved independently from the corresponding structure in the clade that includes *Shansisuchus shansisuchus* and *Chalishevia cothurnata*.

*Peng (1991)* identified a secondary antorbital fenestra in *Guchengosuchus shiguaiensis* based on the morphology of the ascending process of the maxilla, which he described as possessing a distinctly concave anterior margin. Unfortunately, this process has been lost since the original description of the specimen. *Peng (1991)* reconstructed the secondary antorbital fenestra as a slit-like opening between the postnarial process of the premaxilla and the ascending process of the maxilla, with limited or no participation by the nasal (*Peng, 1991*: fig. 2b). The secondary antorbital fenestra of other erythrosuchids (*Shansisuchus shansisuchus* and *Chalishevia cothurnata*) is also formed by the premaxilla and maxilla, but the nasal contributes substantially to the opening. Indeed, the secondary antorbital fenestra of *Shansisuchus shansisuchus* and *Chalishevia cothurnata* is in part a result of the presence of a long, non-articular margin on the anteroventral margin of the nasal, between the articular facets for the reception of the postnarial process of the premaxilla and the ascending process of the maxilla. This non-articular margin is absent in *Guchengosuchus shiguaiensis*, the articular facets of the premaxilla and maxilla being adjacent to each other. If an opening was present between the premaxilla and maxilla, as originally suggested by *Peng (1991)*, this would have been more similar in its position and relationships with the surrounding bones to the subnarial fenestra that has been described for some loricatan pseudosuchians (e.g., *Decuriasuchus quartacolonia*: *França, Langer & Ferigolo, 2013*; *Prestosuchus chiniquensis*: *Roberto-Da-Silva et al., 2016*; but see *Nesbitt & Desojo (2017)* for an alternative interpretation of these openings as the result of deformation during taphonomic processes) and the early dinosaur *Herrerasaurus ischigualastensis* (*Sereno & Novas, 1993*). As a result, we think it is likely that *Guchengosuchus shiguaiensis* did not have a secondary antorbital fenestra homologous with that present in some Middle Triassic erythrosuchids, and we cannot confirm the presence of an opening between the premaxilla and maxilla because the specimen has been damaged since its original description. Despite the uncertainties around this character, however, the phylogenetic position of *Guchengosuchus shiguaiensis* as an erythrosuchid is still relatively well supported, and this species provides useful information about the anatomy of members of this clade during their early evolutionary history.

## APPENDIX

Characters modified from the data matrix of *Ezcurra et al. (2018)*:

*Character 46.* Maxilla-nasal, maxillo-nasal tuberosity, delimiting anteriorly the antorbital fenestra or fossa if it is present: absent (0); present but low, with a gradual transition with the external surface of the anterior process (1), present and high, with a distinct change of slope between it and the external surface of the anterior process (2). ORDERED.

*Character 393.* Scapula, lateral tuber or ridge on the posterior edge: absent (0); present, just dorsal to the glenoid fossa (1); present, around the level of maximum anteroposterior narrowing of the scapular blade (2).

Here, we propose that the ridge on the scapular blade of some erythrosuchids (e.g., *Guchengosuchus shiguaiensis*, *Erythrosuchus africanus*, *Shansisuchus shansisuchus*) is homologous with the tuberosity or ridge present immediately dorsal to the supraglenoid lip in several other archosauriforms (*Gower, 2003*; *Nesbitt, 2011*).

Scorings changed from the data matrix of *Ezcurra et al. (2018)*:

*Character 15* (Secondary antorbital fenestra, immediately anterior to the antorbital fenestra). *Guchengosuchus shiguaiensis* (based on IVPP V8808): changed from (1: present) to (0: absent).

*Character 46* (Maxilla-nasal, maxillo-nasal tuberosity, delimiting anteriorly the antorbital fenestra or fossa if it is present). *Guchengosuchus shiguaiensis*: changed from (0) to (1). *Garjainia prima*, *Erythrosuchus africanus*, *Shansisuchus shansisuchus*, *Chalishevia cothurnata*, and *Batrachotomus kupferzellensis*: changed from (1) to (2) because of the addition of a new character-state. See above for character-states.

*Character 56* (Maxilla, secondary antorbital fossa anteriorly to the antorbital fossa and adjacent to the dorsal margin of the anterior process). *Guchengosuchus shiguaiensis* (based on IVPP V8808): changed from (0: absent) to (-: inapplicable).

*Character 69* (Maxilla, edentulous anterior portion of the ventral margin of the bone). *Guchengosuchus shiguaiensis* (based on IVPP V8808): changed from (0: absent) to (?: missing data).

*Character 393* (Scapula, lateral tuber or ridge on the posterior edge). *Guchengosuchus shiguaiensis* (based on *Peng, 1991*), *Erythrosuchus africanus* (based on *Gower, 2003*), *Shansisuchus shansisuchus* (based on *Young, 1964*): changed from (0) to (2) because of the addition of a new character-state. See above for character-states.

*Character 652* (Articular, medial surface). "*Chasmatosaurus*" *yuani* (based on IVPP V4067), *Guchengosuchus shiguaiensis* (based on IVPP V8808), *Proterosuchus alexanderi* (based on NM QR 1484), *Proterosuchus fergusi* (based on BSPG 1934 VIII 514): changed from (0: without dorsomedial projection posterior to the glenoid fossa) to (1: with dorsomedial projection separated from glenoid fossa by a clear concave surface).

## INSTITUTIONAL ABBREVIATIONS

| | |
|---|---|
| **BP** | Evolutionary Studies Institute (formerly Bernard Price Institute for Palaeontological Research), University of the Witwatersrand, Johannesburg, South Africa |
| **BSPG** | Bayerische Staatssammlung für Paläontologie und Geologie, Munich, Germany |
| **GHG** | Geological Survey, Pretoria, South Africa |
| **GMB** | Geological Institute, Beijing, China |
| **ISIR** | Indian Statistical Institute, Kolkata, India |
| **IVPP** | Institute of Vertebrate Paleontology and Paleoanthropology, Beijing, China |
| **MCNAM** | Museo de Ciencias Naturales y Antropológicas de Mendoza (J. C. Moyano), Mendoza, Argentina |
| **MCZ** | Museum of Comparative Zoology, Cambridge, USA |
| **NHMUK PV** | The Natural History Museum, Palaeontology Vertebrates, London, UK |
| **NM QR** | National Museum, Bloemfontein, South Africa |
| **PIN** | Borissiak Paleontological Institute of the Russian Academy of Sciences, Moscow, Russia |
| **PULR** | Paleontología, Universidad Nacional de La Rioja, La Rioja, Argentina |
| **PVL** | Paleontología de Vertebrados, Instituto "Miguel Lillo," San Miguel de Tucumán, Argentina |
| **QM** | Queensland Museum, Brisbane, Queensland, Australia |
| **RC** | Rubidge Collection, Wellwood, Graaff-Reinet, South Africa |
| **SAM-PK** | Iziko South African Museum, Cape Town, South Africa |
| **TM** | Ditsong National Museum of Natural History (formerly Transvaal Museum), Pretoria, South Africa |
| **UMZC** | University Museum of Zoology, Cambridge, UK |
| **UNIPAMPA** | Universidade Federal do Pampa, São Gabriel, Brazil |
| **UTGD** | School of Earth Sciences, University of Tasmania, Hobart, Australia |
| **ZPAL** | Institute of Paleobiology of the Polish Academy of Sciences, Warsaw, Poland. |

## ACKNOWLEDGEMENTS

We thank Michael Parrish and Felipe Pinheiro for helpful review comments, and Andrew Farke for editorial comments. Michael Parrish provided photographs of some of the *Guchengosuchus* material as preserved during his 1990 visit to Beijing.

### Funding

Richard J Butler, Martin D Ezcurra and Roland B Sookias were supported by a DFG Emmy Noether Programme award (BU 2587/3-1) and a Marie Curie Career Integration Grant (grant number 630123). Corwin Sullivan was supported by the Natural Sciences and

Engineering Research Council of Canada (Discovery Grant RGPIN-2017-06246), and University of Alberta start up funds. Corwin Sullivan and Jun Liu were supported by the National Natural Science Foundation of China (grant number 41472017, 41661134047). Roland B Sookias was also supported by a Humboldt Research Fellowship from the Alexander von Humboldt Foundation. The funders had no role in study design, data collection and analysis, decision to publish, or preparation of the manuscript.

## Grant Disclosures

The following grant information was disclosed by the authors:
DFG Emmy Noether Programme award: BU 2587/3-1.
Marie Curie Career Integration Grant: 630123.
Natural Sciences and Engineering Research Council of Canada: Discovery Grant RGPIN-2017-06246.
University of Alberta start up funds.
National Natural Science Foundation of China: 41472017, 41661134047.
Humboldt Research Fellowship from the Alexander von Humboldt Foundation.

## Competing Interests

The authors declare that they have no competing interests.

## Author Contributions

- Richard J. Butler conceived and designed the experiments, performed the experiments, analyzed the data, contributed reagents/materials/analysis tools, prepared figures and/or tables, authored or reviewed drafts of the paper, approved the final draft.
- Martín D. Ezcurra conceived and designed the experiments, performed the experiments, analyzed the data, contributed reagents/materials/analysis tools, prepared figures and/or tables, authored or reviewed drafts of the paper, approved the final draft.
- Jun Liu conceived and designed the experiments, performed the experiments, analyzed the data, contributed reagents/materials/analysis tools, authored or reviewed drafts of the paper, approved the final draft.
- Roland B. Sookias conceived and designed the experiments, performed the experiments, analyzed the data, contributed reagents/materials/analysis tools, prepared figures and/or tables, authored or reviewed drafts of the paper, approved the final draft.
- Corwin Sullivan conceived and designed the experiments, performed the experiments, analyzed the data, contributed reagents/materials/analysis tools, authored or reviewed drafts of the paper, approved the final draft.

## Data Availability

   Raw data are available in the figures (anatomical data) and Supplementary Material (additional figures and the phylogenetic data matrix).

## Supplemental Information

Supplemental information for this article can be found online at http://dx.doi.org/10.7717/peerj.6435#supplemental-information.

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
