# Peer review of "The anatomy and phylogenetic position of the erythrosuchid archosauriform Guchengosuchus shiguaiensis from the earliest Middle Triassic of China"

_PeerJ, doi:10.7717/peerj.6435_

## Round 0.1 · original submission · Minor Revisions

Overall, the manuscript and its analyses are well presented and well executed. Figure quality is high, and this in particular will be very useful for the research community.

There are a few areas where I feel that additional measurements (e.g., centrum dimensions) would be helpful, but I do not consider the point mandatory for revision.

The phylogenetic data matrix (in TNT or NEXUS format) and character list should be presented as supplemental material, to allow easier reanalysis of the data by the reader, and to more easily make the paper a standalone unit (as noted by the reviewers).

·

Basic reporting

Butler and colleagues provide a competent, detailed description of the erythrosuchid Guchengosuchus shiguaiensis, a yet poorly known member of an intriguing clade of Early/Middle Triassic predators. Although I’m not a native English speaker myself and probably not the best person to evaluate this, the manuscript seems to be written in clear, professional English and, apart from some minor misspelling of anatomical terms (see below), I did not detect any language errors. The authors provide a relevant, well referenced background, being fully successful in contextualize their objectives. The authors provide high-quality photographs and one drawing of the specimens, and all relevant structures are correctly labeled and described within the text.

Experimental design

The experimental design (that is, the phylogenetic protocol) is adequate, and the methodology well described (but see below).

Validity of the findings

Although the data is meaningful and the phylogenetic analysis is sound, I have some concerns about the availability of data (see below). All the conclusions are well supported by methodological background and presented results.

Additional comments

I have some minor to moderate concerns that, although not compromising the quality of the manuscript, should probably be addressed by the authors:

1. Raw data availability: as the phylogenetic protocol was based on a still unpublished data matrix (Ezcurra et al., in press), reproducibility of experiments is compromised. Although this is maybe unlikely to happen, in case the manuscript here evaluated is published before Ezcurra et al., the raw data should be provided.
2. Despite the description being exceptionally detailed, some comparisons are missing. I suggest the authors to improve the text by adding detailed comparisons between Guchengosuchus dorsal vertebrae and ribs with the same elements of other archosauromorphs.
3. Figures: “postzygodiapophyseal lamina” and “prezygodiapophyseal lamina” are misspelled in Figure 13 label. Also correct similar terms in text lines 707 and 711. Also in figure 13, the label pp.af is not indicated on the photographs.
4. Keywords: I suggest the authors to not repeat words from the title as keywords.

Line 52: Ezcurra et al., 2013; Ezcurra et al., in press.
Lines 53-54: please cite, for instance, Ochev (1958) and Gower & Sennikov (2000) to refer to Russian erythrosuchids.
Lines 180-182: As the anterior process of the maxilla is broken, maybe it would be precipitate to infer that this structure is ‘anteroposteriorly short’.
Lines 198-200: This lateral thickening of the ascending process is apparently interpreted as an earlier stage of the transformation series that led to the development of the characteristic maxillo-nasal tuberosity of later forms. It would be interesting to see this scored as a multistate character in the data matrix (in this case, Ezcurra, 2016 character 46).
Lines 289-292: Teyujagua paradoxa has marginal teeth that appear superficially to be fully thecodont, meaning that teeth are deeply inserted into alveoli and tooth crowns are apparently separated from bone walls, without striae. CT scans, however, showed us that, deep inside the alveoli, the teeth are ankylosed to surrounding bone. Could this be also the case for Garjainia and Erythrosuchus?
Lines 676-677: Could the transversal compression of the vertebrae be exaggerating the oval outline of the articular surfaces of centra?
Line 717: Could you please provide examples of the presence of spinoprezygapophyseal laminae in other archosauromorphs?

·

Basic reporting

The authors have done a very good job of redescrbing the holotype material of the basal archosauriform Guchengosuchus shiguaiensis and discussing its phylogenetic relationships to the erythrosuchids. Redescription of this specimen is important because it greatly improves on the detail of the figures included with the original description of the material by Peng (1991) and because of the damage and loss of material that the specimen has sustained since its original description. The damage sustained to the G. Shiguaiensis holotype and the loss of the holotype of the other Shanxi taxon Fugusuchus hejiapensis greatly limits the ability to reinterpret the basal states of erythrosuchids based on subsequent analyses. The paper is clearly written, the descriptions appropriate, and the new figures represent a dramatic improvement on the figures from Peng (1991).

Lines 577-578 – remove ‘for’ or complete the phrase.

Experimental design

The authors split their morphological descriptions between new descriptions of the holotype material based on examination by the authors and attempts to re-interpret the photographs, drawings, and descriptions from Peng (1991) for the material that is now either missing or damaged. The new descriptions are thorough and the characters well illustrated and labeled. However, it appears that some of the information based on Peng’s (1991) illustrations and descriptions may be interpreted beyond the available resources, most specifically in lines 853-864 discussing a putative tuberosity on the scapula which is barely visible in Peng's figures and in the discussion of the possible second ant orbital fenestra (lines 945-971), the existence of which cannot be evaluated fully without the complete ascending process of the maxilla and, ideally, the posterior margin of the premaxilla (an element not known in Guchengosuchus, although Peng also speculated on the fenestra's presence).

Validity of the findings

The phylogenetic conclusions of the authors seem consistent with the material as described. However, it would be helpful to readers if more information was provided in the article and supplemental materials so that readers could interpret the data without having to refer to other analyses done by one or more of the co-authors. Specifically, Appendix A should describe the characters being modified from the Ezcurra et al (in press) character list, not simply list the numbers in the original analysis. In the discussion of the matrix in Materials and Methods (lines 89-95, the authors mention that the data set originally published by Excurra (2016) was used as a basis for the study, but as modified by findings in six subsequent studies. With so much modification, the matrix used should either be provided as an appendix or made available as a link in an online source like treebase. Also, the assertions in lines 926-928 ( “The phylogenetic relationships of Erythrosuchidae found in our analysis are congruent with those previously recovered by Ezcurra (2016) and Ezcurra et al. (in press). The interrelationships within Erythrosuchidae are relatively robust and do not generate substantial ghost lineages.”) should be expanded in light of the extensive tree analysis and manipulation described in lines 92-114. Even though the focus of the paper is relationships among erythrosuchids, some outgroups should be included in Figure 20 or an additional cladogram, along with a more detailed discussion of how the altered character states affect the larger phylogenetic results.

Additional comments

Overall, an important addition to the anatomical and phylogenetic knowledge of the Erythrosuchidae. I have photos of the Guchengosaurus material taken during my 1990 visit to Beijing that show the specimen in its original state that I would be happy to share with the authors if I can get the slides scanned into a digital format. Incidentally, I was shown the material during that visit but asked not to include it in my analysis because of Peng's pending publication, which is why it was not included in my 1992 paper on erythrosuchid systematics.

---

## Round 0.2 · accepted · Accept

Thank you for your close attention to the comments from the reviewers.

#